# Piezoelectric Direct Discharge: Devices and Applications

**Dariusz Korzec * , Florian Hoppenthaler and Stefan Nettesheim**

Relyon Plasma GmbH, Osterhofener Straße 6, 93055 Rgensburg, Germany;
f.hoppenthaler@relyon-plasma.com (F.H.); s.nettesheime@relyon-plasma.com (S.N.)
* Correspondence: d.korzec@relyon-plasma.com

**Abstract:** The piezoelectric direct discharge (PDD) is a comparatively new type of atmospheric pressure gaseous discharge for production of cold plasma. The generation of such discharge is possible using the piezoelectric cold plasma generator (PCPG) which comprises the resonant piezoelectric transformer (RPT) with voltage transformation ratio of more than 1000, allowing for reaching the output voltage >10 kV at low input voltage, typically below 25 V. As ionization gas for the PDD, either air or various gas mixtures are used. Despite some similarities with corona discharge and dielectric barrier discharge, the ignition of micro-discharges directly at the ceramic surface makes PDD unique in its physics and application potential. The PDD is used directly, in open discharge structures, mainly for treatment of electrically nonconducting surfaces. It is also applied as a plasma bridge to bias different excitation electrodes, applicable for a broad range of substrate materials. In this review, the most important architectures of the PDD based discharges are presented. The operation principle, the main operational characteristics and the example applications, exploiting the specific properties of the discharge configurations, are discussed. Due to the moderate power achievable by PCPG, of typically less than 10 W, the focus of this review is on applications involving thermally sensitive materials, including food, organic tissues, and liquids.

**Keywords:** atmospheric plasma; resonant piezoelectric transformer; piezoelectric direct discharge; ozone; surface activation; disinfection

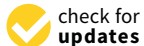



## 1. Introduction

The low temperature or cold atmospheric pressure plasmas (APP) are versatile tools in a large number of human activities [1–4]. Their applications are ranging from improvement of industrial production processes [5], to numerous applications in biology, genetics, and medicine [6,7]. The increasing demand on a compact, affordable, and flexible plasma tools motivated the development of a new family of piezoelectric cold plasma generators (PCPG) [8] based on the resonant piezoelectric transformer (RPT) principle [9]. The use of PCPG to produce the piezoelectric direct discharge (PDD) [10] is the focus point of this review. The operation power range of 3 to 10 W makes PCPGs especially suitable for implementation in compact desktop instruments or in handheld atmospheric pressure plasma jets (APPJ) [11]. The low temperature of the produced plasma gases, only a few K higher than the ambient temperature, makes the treatment of fruits, seeds, and tissues feasible. The high achievable ozone concentration allows for application for disinfection and sterilization. These are only a few of the strongly diverging application field examples. In this review, different configurations of PCPG driven devices are classified, and their operation principles are explained. The suitability of these configurations for specific classes of applications is discussed and illustrated with practical application examples. The authors hope that this review will inspire completely new fascinating approaches and application fields, not yet revealed.

## 2. PDD Generation

### 2.1. PCPG Development

The voltage needed for ignition of the PDD is produced by the resonant piezoelectric transformer (RPT). In the next sections, the way from RPT to PCPG and the basics related to its use for PDD production are shown.

#### 2.1.1. High Voltage RPTs

The first RPT consisting of two quartz crystals is described in [12]. A single quartz crystal RPT is disclosed in [13]. A large step in the direction of high power, high voltage transformation ratio RPT is the invention of Rosen [14]. The Rosen type RPT consists typically of an elongated block made of piezo-electric, preferentially ferroelectric material, having a resonant mode of vibration, consisting of two zones, input zone polarized transversally to the vibration biased by small voltage and the output zone polarized longitudinally to the vibration, producing a high voltage on its tip. It is a compact and versatile tool for generation of a high voltage in a kHz range [15]. This invention was a starting point of a very diversified research and development, reviewed, e.g., in [9]. Performance of an RPT is drastically improved by application of materials belonging to PZT ($PbZr_{1-x}Ti_xO_3$ = lead zirconate titanate) ceramics [16,17]. Numerous RPT applications for power electronics are reported [18]. Especially important for plasma applications are the high power high voltage RPTs. An optimized multilayer structure enables the RPT with a maximum power transfer [19] and a high voltage transformation ratio [20,21]. Further efforts are made to develop a lead-free RPT. An example of an RPT device based on Lithium niobate, instead of PZT, is the ionic wind generator proposed in [22,23]. However, currently, no alternative material can reach the performance of PZT in the high voltage RPTs.

#### 2.1.2. Powering Low Pressure Discharges

For a long time, the RPTs were a lightweight, high efficiency replacement of conventional magnetic transformers [9]. As such, the RPT was very successful for a discharge ignition in cold cathode fluorescent lamps (CCFL) in flat screen backlight modules [24,25]. Typical examples are the backlight sources of laptops and hand-held devices [26]. The powering of neon lamps by RPTs was also shown [27]. The RPTs developed for powering of CCFLs are strong enough to be applied as a plasma source [28]. The applications of RPT for generation of a low pressure discharges are described in [29,30].

#### 2.1.3. PDD in Noble Gas

From this point of development, it was only a small step to provide RPTs with output voltage sufficient to ignite atmospheric pressure discharge in noble gases [31]. It was possible in noble gases but not in an atmospheric air because the break-down voltage in noble gases is much lower than in atmospheric air [32,33]. In those early devices, frequently, the RPT was not used as a discharge electrode, but worked as a transformer biasing a metal electrode being in contact with plasma [34]. The piezobrush® PZ1 [35] (see the instrument first from the left in Figure 1), the first commercial device generating the PDD in helium, is based on RPT working with the resonant frequency of 135 kHz, input power of 20 W, and the output voltage of 1.8 kV. The piezobrush® PZ1 is a useful tool for surface activation before processing steps such as: gluing, painting, printing, casting, foaming, coating, or siliconizing. Specifically, an increase of wettability, printability, or adhesion on polymers [36,37] and ceramics [38,39] are reported. The activation allowed for grafting of different chemically active films, e.g., silica capillary nozzles are treated 1 min before dipping into a silanisation solution (5% dimethyldichlorosilane in heptane) [40] or grafting of polymer brushes on the plasma treated ETFE (ethylene tetrafluoroethylene), PP, and PE surfaces allowing for better wettability of these surfaces [41], or improvement of surface properties of zirconia for dental restoration [38].

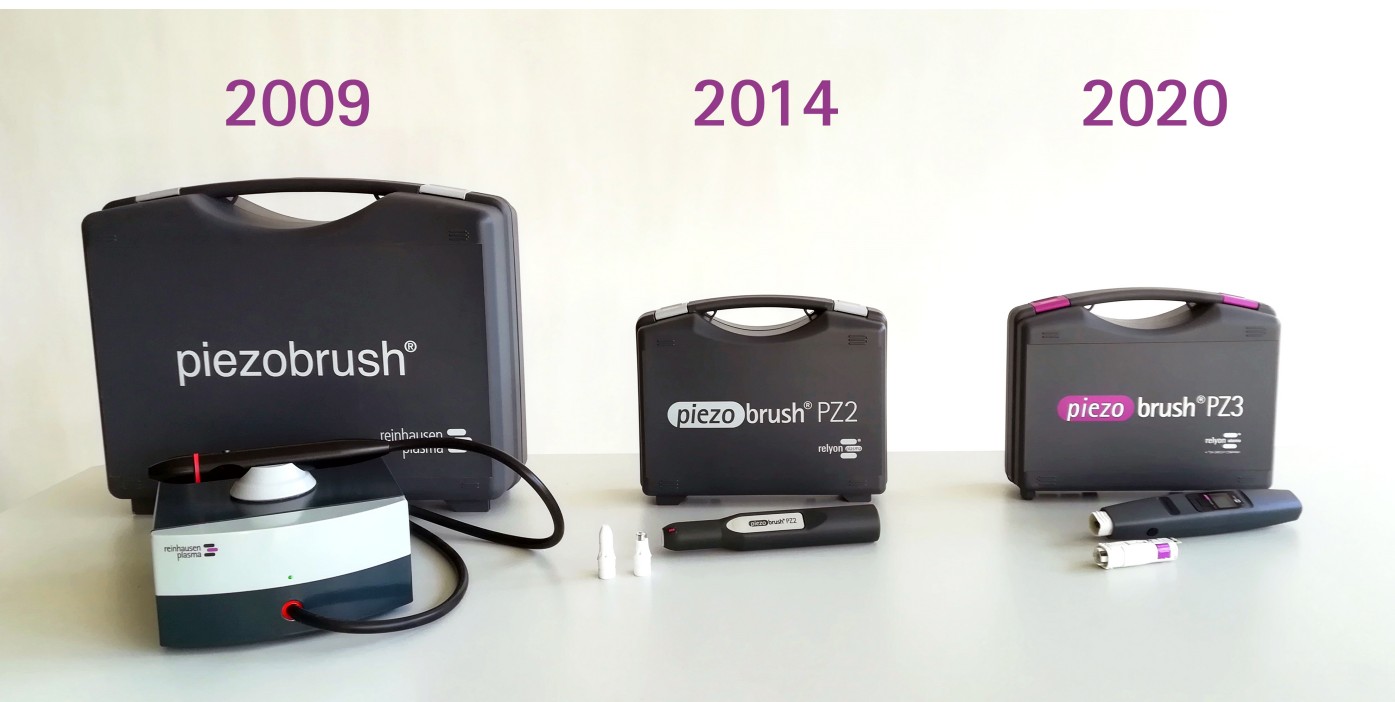

**Figure 1.** Three generations of the PDD based piezo-brushes (Courtesy of Relyon Plasma GmbH).

### 2.1.4. PDD in Air

Recently, PCPGs based on RPT with high transformation ratio (up to 1000) were developed, allowing generation of the atmospheric pressure discharge in air [8]. They are constructed as a two-zone cuboid-shaped Rosen type, PZT multilayer RPT. Two of them, CeraPlas™ F [42] and CeraPlas™ HF [43], are available commercially. Their typical parameters are summarized in Table 1. The main difference between them is the operation frequency. The 2nd harmonic frequency of a resonant longitudinal mechanical oscillation is used for the excitation of the PCPG. This frequency is closely related to the sizes of the PCPG block and to the material properties, e.g., speed of sound [44]. Most of the results shown in this review are collected using CeraPlas™ F. This PCPG is also the core component of the hand-held plasma brushes piezobrush® PZ2 and piezobrush® PZ3 shown in Figure 1.

**Table 1.** The typical parameter of two PCPGs.

| PCPG Type | CeraPlas™ F [45] | CeraPlas™ HF [43] |
|---|---|---|
| operating frequency [kHz] | 50 | 82 |
| weight [g] | 8.0 | 3.8 |
| length × widht × thickness [mm] | 72 × 6 × 2.8 | 45 × 4.0 × 2.8 |
| piezoelectric material | PZT | PZT |
| maximum operating power [W] | 8.0 | 4.0 |
| input capacity $C_{in}$ [ μF ] | ∼2.0 | ∼0.74 |
| output capacity $C_{out}$ [ pF ] | ∼3.0 | ∼2.1 |

### 2.2. The PCPG Operation Principle

The task of the PCPG is to generate a high AC voltage, typically over 10 kV. The electric field at the PCPG tip causes an ionization processes in the surrounding gas and the initiation of micro-discharges [46]. The voltage transformation is achieved at resonant frequency due to the formation of a standing acoustic wave which transforms the low voltage from the input side to a high voltage at the mechanically coupled output side. The following

sections show how the generation of high voltage and sustaining plasma can be achieved by the piezoelectric principle and by control of the PCPG.

2.2.1. Piezoelectric Transformer Principle

The PCPG converts, similar to the common magnetic transformers, the electric AC input signal with low voltage and high current to the output signal with high voltage and low current. Thereby, its performance is characterized by the voltage transformation ratio defined as the ratio of the output voltage to the input voltage. Essential for operation of the PCPG are two physical effects: the direct piezoelectric effect and the indirect (converse) piezoelectric effect [47]. The direct piezoelectric effect causes the increase of the polarization strength and, consequently, the surface charge density on two sides of a piezoelectric material, when it is subjected to the external mechanical stress. This effect is especially strong if the mechanical force is applied in the direction of the polarization vector in a pre-polarized ferroelectric material. The converse piezoelectric effect occurs if a mechanical deformation of the piezoelectric material follows the voltage applied on this material. In addition, this effect is strongest in pre-polarized ferroelectric material blocks when the electric field is applied parallel to the vector of polarization.

The idea of the PT is to combine these two effects in a two-zone material block. In the primary zone, a small voltage is applied to induce its geometrical deformation (converse piezoelectric effect). The primary zone is mechanically connected to the secondary zone, in which the mechanical deformation of the first block is causing the mechanical stress, resulting in the generation of a high voltage. A specific technical solution is a two-zone cuboid-shaped block.

Typically, the primary zone is polarized perpendicular to the cuboid long axis, and, in the same way, the voltage is applied. The secondary zone is polarized parallel to the long axis of the cuboid, causing generation of a high voltage due to the axial force from the first block. Such mechanical deformation of the input zone of the PT can be reached at much lower input voltage if a multilayer structure is applied [14]. A further measure to increase the voltage transformation ratio is using for the electric excitation signal a frequency close to the mechanical resonance frequency. By choosing the 2nd harmonic vibration mode as an operation mode, it is possible to contact and mount the piezoelectric transformer at the vibration nodal points without disturbing its mechanical movement. These points are also used for mechanical fixing the device.

2.2.2. PCPG as Resonator

The PCPG, as any block of material with regular shapes, can oscillate with its own resonance frequency determined by its geometrical sizes and the material properties (speed of sound) [48]. The behavior of an RPT, considered as a damped harmonic oscillator [49], can be described using a resonance curve (see Figure 2) showing the dependence of the voltage transformation ratio on the excitation frequency. The voltage transformation ratio has its maximum value for resonance frequency. The decay of not sustained oscillation in a resonator is described by damping ratio $\zeta$, which is the reciprocal of the oscillation quality factor. The smaller the full width at a half maximum of the resonant curve, the lower the damping ratio. At damping ratio values higher than 0.3, no resonant oscillation is possible (overdamped oscillation) [49]. For very small values of the damping ratio, the value of the voltage transformation ratio and consequently the oscillation amplitude at resonance frequency is high. Figure 2 illustrates the influence of the damping on the resonance curve. With increasing damping ratio, in the shown example from 0.001 to 0.0025, the maximum voltage transformation ratio decreases. At the same time, the full width at half maximum increases and the resonance frequency decreases. The main reasons for damping of the PCPG oscillations are:

- mechanical losses in the PZT material,
- friction of the holder system and the electric connections,
- electrostatic losses,

- capacitive coupling, and
- ohmic losses due to plasma.

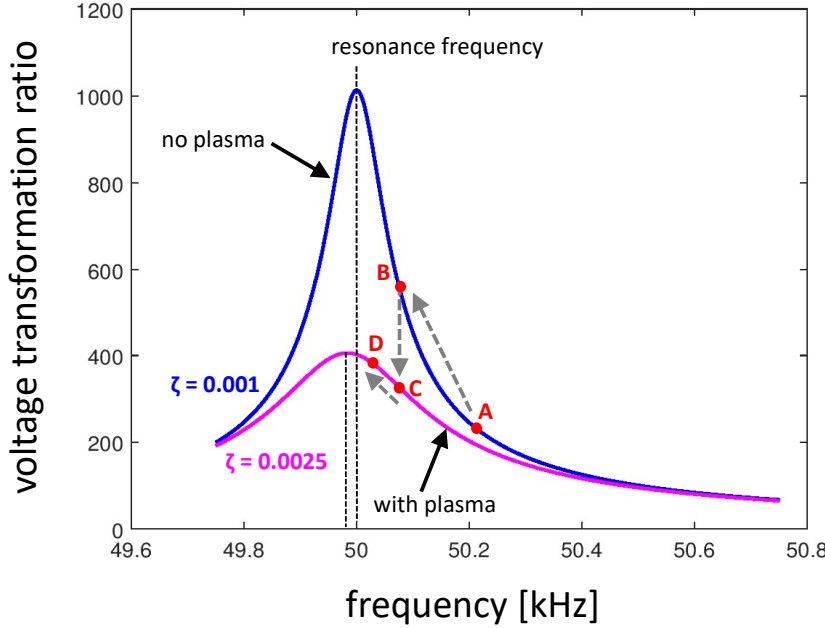

**Figure 2.** Trajectory of the working point of the RPT on the resonant curves during plasma ignition.

### 2.2.3. Excitation Frequency Control

It seems easy to excite the PCPG with a frequency assuring the high voltage transformation ratio by fixing its value close to the resonant frequency. However, no stable operation can be established with constant excitation frequency. Different factors cause variation in the resonant frequency of the device, e.g., the PCPG ambient temperature, the heating during operation, and the electrical loading [50]. The frequency is changed also to control the input power, which can't be controlled directly. The power level is controlled by setting of input voltage, input current, and the excitation frequency [51]. To sustain a stable operation, a fitting feedback control was implemented. When the phase shift of the PCPG input current relative to the phase of the input voltage is zero, the voltage transformation ratio of the system is at its maximum [52]. Any change in the operating frequency is detected and adjusted to operate at maximum efficiency again [53].

### 2.2.4. Plasma Ignition

Figure 2 shows the operational interaction between the plasma load and the PCPG. Before the plasma ignition, the electric load of the PCPG is a small capacity, causing a weak damping. The operating point of the PCPG will be on the blue resonance curve in Figure 2 representing the low damping conditions. Since the exact plasma ignition voltage and operating frequency are not known, the starting point of the frequency control is on the right side of the resonant curve in the region of a low voltage transformation ratio. This is shown as point A. As the operating frequency is decreased, the PCPG voltage transformation ratio moves up the "no plasma" line until the plasma ignition voltage is reached. This happens in operating point B. After plasma ignition, the damping ratio increases abruptly. The ignition of plasma causes reduction of the resonant frequency and the voltage transformation ratio of the PCPG [54]. The resonance curve "no plasma" is not valid anymore. The operating point jumps to the resonance curve "with plasma" valid for stronger damping and reaches the operating point C. Subsequently, the operating point moves from point C to point D, causing an increase of voltage transformation ratio and, consequently, plasma intensity. The operating frequency of the point D is reduced this way so that the input power of the PCPG can reach the set value. In Figure 2, the plasma operation is represented by the single curve. In reality, the plasma loaded resonance

curve is permanently changing, adopting its shape to the non-constant damping ratio. The firmware used to control the PCPG keeps the operating point on the right side of the resonance curve to avoid control instabilities.

### 2.2.5. Sustaining the PDD

The periodic input voltage excites the oscillation of the PCPG not abruptly, but within a number of oscillation cycles. Depending on the mechanical vibration quality factor, several tens to several hundred oscillation cycles are needed to reach the output voltage value high enough for discharge ignition. During a single half-period of the voltage excitation, the multiple micro-discharges can be generated from restricted areas of the PZT surface. The output capacity plays an important role in prediction of the electric behavior of the PCPG as it consists of parallel connection of the capacity of a non-loaded PCPG (see $C_{out}$ in Table 1) and capacity of the capacitive PCPG load $C_{load}$. The last one varies if plasma ignites. This capacity is storing and delivering the charge needed for the evolution of the micro-discharges. The plasma ignition causes a decrease of the output impedance and, consequently, a reduction of the resonant frequency, increase of the full width at half maximum of the resonance curve, and decrease of the voltage transformation ratio of the PCPG.

### 2.2.6. Fixing and Packaging

One of the critical points in the construction of the PCPG based devices is their fixing and attachment of electric connections. The electric connections are made frequently of a thin braided wire, in order to reduce the damping of the RPT oscillation. For moderate voltages, as typically used for power electronics' applications, different types of casting are commonly used to insulate and mechanically stabilize the RPT. However, in the case of PCPG, the electric fields at the high voltage side are so high that it is difficult to find a casting material withstanding this electrostatic load and, at the same time, being a sufficiently good heat conductor to avoid the heat build-up. The solution applied for the PCPG is to let it open for air cooling and fixing it only in the points of minimum mechanical vibration. By excitation of the PCPG with its 2nd harmonic frequency, such points are positioned at 1/4th and 3/4th of its length. Figure 3a shows the PCPG suspension on two elastomer holders realized in the piezobrush® PZ2. The solder connections in the 1/4th position are depicted. The round holes in the holders are for assuring the cooling gas flow along the PCPG .

Due to the mechanical standing wave, the maximum electric field is generated on the tip of the PCPG. However, in some cases, the field along the edges of the HV side of the PCPG is strong enough to ignite a parasitic discharge. This can happen, for example, if some electrically conducting objects are approaching the PCPG HV edges, the PCPG is working under reduced pressure conditions or in a gas mixture with low break-down voltage (noble gas mixtures). To avoid such parasitic discharges, the rounding of the edges and insulating coatings and covers are used [55] as shown in Figure 3b.

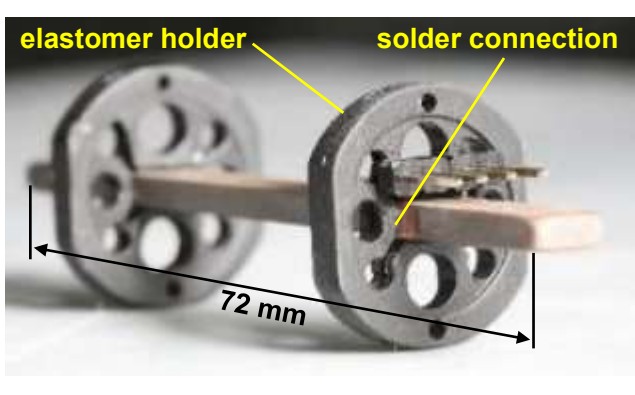

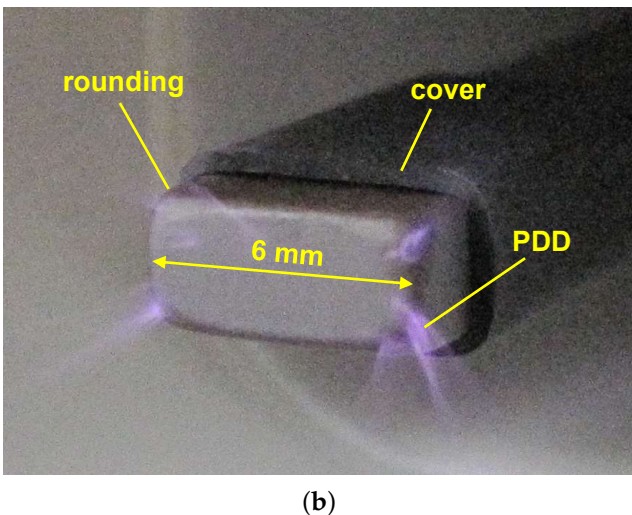

(**a**) (**b**)

**Figure 3.** The PCPG mechanical features. (**a**) suspension of the PCPG prepared for assembling in the piezobrush® PZ2 (courtesy of Relyon Plasma GmbH); (**b**) the PDD at the front face of the PCPG; and the features to suppress the parasitic discharges are depicted.

## 3. PDD Evaluation Methods

In principle, all plasma diagnostics methods used for APP [56,57], especially for APPJ [58–61] can be applied for the characterization of the PDD. Additionally, some techniques used for characterization of RTP operation [62] can also be useful. The focus here is on methods, which were applied to evaluate PDD efficiency in practical applications, such as hand-held devices. These are the capacitive probe measurements explained in Section 3.1, the determination of ozone production rate presented in Section 3.2, the evaluation of the activation surface area described in Section 3.3, and the thermal analysis by use of IR recording discussed in Section 3.4.

### 3.1. Electrical Characterization

The most important performance parameter of an RPT is the voltage transformation ratio. To determine it, the input and output voltage amplitudes are needed, but it is not possible to measure the output voltage of the PCPG directly, because it doesn't have any output electrode. The alternative is to evaluate the signals induced by the electric field of the PCPG. Doing so, it is important not to influence the PCPG by the measurement. One approach is to measure the electric fields in the proximity of the PCPG output by the use of methods not involving electrically conducting objects, for example using an electro-optic effect [63]. The application of the electro-optical Pockels effect in a CdTe crystal sensor is described in [64].

If electric sensors are used [65,66], they should either have very high impedance, like in the measurements with a matrix of capacitive high impedance probes shown in [67], or the sensor should be positioned at a sufficiently large distance. The technique based on a large area capacitive probe placed in a safe (not influencing the PDD) distance from the PCPG is described in [68]. The voltage from this probe allows for the determination of both the voltage amplitude of the PCPG output induced by piezoelectric oscillation, and the number of the micro-discharges per oscillation period of the PCPG.

Figure 4a,b shows the output voltage of the CeraPlas™ F measured by use of the capacitive probe, as described in detail in [11]. Each of these curves can be considered as an overlap of a more-or-less sinusoidal, periodic curve and short non-periodic pulses at the maximum and minimum of the periodic curve, representing the electric reaction of the probe on micro-discharges. The period of the sinusoidal component corresponds to the frequency of the 2nd harmonic of the resonant CeraPlas™ F oscillation of 50 kHz.

The voltage for two periods of the CeraPlas™ F oscillation is shown. In the examples, the amplitude of the sinusoidal voltage is twice as high for 8 W than for 4 W CeraPlas™ F power. This periodic signal can be observed even if no plasma is present at the CeraPlas™ F tip. The short periodic pulses are influenced by the CeraPlas™ F input power. It can be seen that the number of anodic (close to the maximum of sinusoidal voltage) micro-discharges is higher for 8 W than for 4 W. The micro-discharge pulses are present only when plasma is generated. The pattern of these pulses is random; therefore, for diagnostic purposes, the mean value of many cycles of the PCPG oscillations must be calculated.

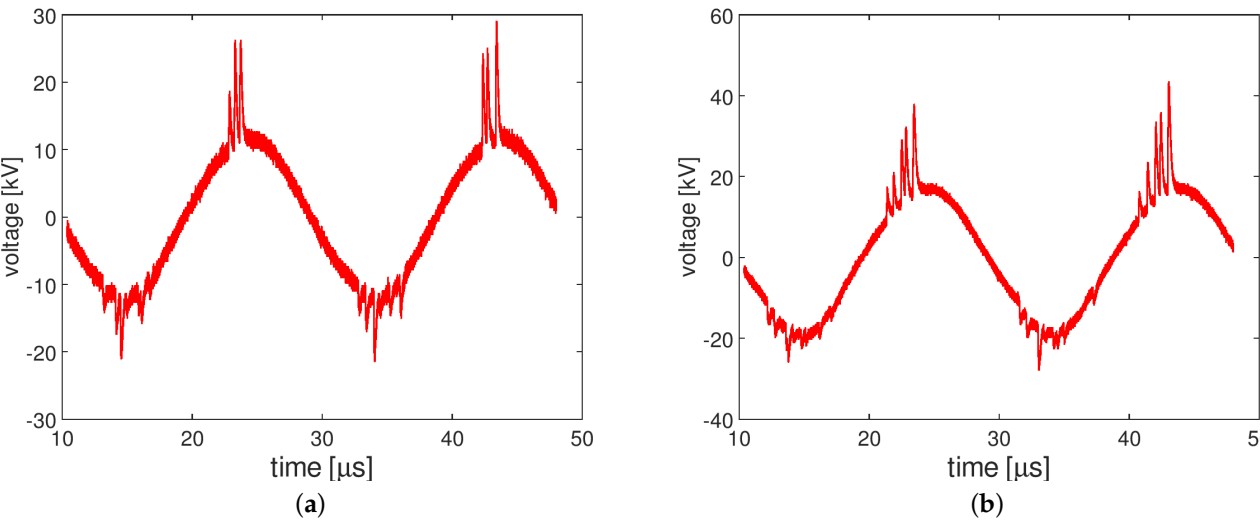

**Figure 4.** CeraPlas™ F output voltage determined by capacitive probe for (**a**) 4 W (**b**) 8 W.

The signals at the minimum and maximum of the periodic curve are different. Typically, the peaks occurring when voltage is positive (anodic peaks) are stronger than the peaks present when voltage is negative (cathodic peaks). The difference in physics of the positive and negative streamers occurring in the early phase of the micro-discharge development is responsible for such asymmetry [69]. A positive streamer needs only half as large an electric field as a negative streamer for its propagation. It also reaches much larger dimensions than the negative one.

Since the lifetime of the micro-discharge corresponds to frequencies in the GHz range, the attenuation and the phase angle of such signal measured by the capacitive probe is different when compared with 50 kHz of piezoelectric oscillation. Because the attenuation of the micro-discharge signal is several orders of magnitude weaker than for the piezo-oscillation signal, the magnitudes related to the micro-discharges and the piezo-oscillations in Figure 4 cannot be directly compared. In addition, the phase position of the micro-discharge pulses in this plot does not represent the factual one due to the frequency dependence of the phase angle. In addition, the duration of the voltage pulses does not mirror the duration of the micro-discharges due to dispersion and high frequency RLC oscillations in the electric circuit [70]. The only information, which can be extracted directly from the micro-discharge responses, is their number per cycle. This value can be used for evaluation of the PCPG efficiency.

The higher the amplitude of the sinusoidal voltage signal, the larger the number and the height of the micro-discharge signals. Since the micro-discharges are responsible for generation of chemically active species, their number per excitation cycle correlates with the ozone production rate and activation efficiency (see Sections 4.6 and 5.1, respectively).

### 3.2. Ozone Concentration Measurement

The gaseous discharge in the air produces a large number of chemically active and excited species [71]. They play a crucial role in all chemical reactions between plasma and

the treated surface. Therefore, it is advantageous to maximize their concentrations. Most of them have short leaving, in the ns to μs range, and, consequently, it is quite difficult for quantitative analysis. One comparatively stable product of the cold air plasma is ozone with a typical half-life time (The half-life time is defined as a time that goes by until the half of the starting ozone amount in a closed space is destructed.) in the range of hours [72]. There exist several measurement techniques for determination of the ozone concentration in the gas phase which can be easily implemented in the lab [73]. Thus, it is possible to use this value as an evaluation parameter for plasma generators. Systematic measurements of ozone concentration have been conducted for the performance evaluation of the PCPG.

Due to high accuracy in a broad range of ozone concentration, the UV absorption spectroscopy is frequently used [74,75]. Various companies offer desktop instruments allowing measurements based on such principles. One of them is ECO Sensors, Inc.f: Santa Fe, New Mexico 87505 USA. For the ozone concentration measurements presented in this work, their Ozone Analyzer Model UV-100 is used, allowing the Ozone concentration measurement in the range from 0.01 to 1000 ppm (volume). Its use for measurement in gas flow is described in Section 4.1.

### 3.3. Activation Area Evaluation

Two parameters can be used for evaluation of the polymer surface treatment result: (i) the surface energy reached after a treatment and (ii) the achievable activation rate, defined as the area of the surface, at which the required surface energy is exceeded, divided by the treatment time.

The standard method for surface energy measurement is the droplet test, or more specifically the contact angle determination, conducted with at least two test liquids [76,77]. If the tests are conducted with the same material all the time, the contact angle measurement with a single liquid, typically DI water, is also very informative [78]. Such method is used in Sections 5.2.1, 6.5, and 7.2.

Another method is using the test inks calibrated for different surface energies, typically in steps of 1 mN/m. When using the PDD based APPJ for surface activation, the maximum surface energies are reached in a very short time, and it is difficult to use these values for quantitative characterization of the APPJ performance. Much more suitable for quantitative evaluation is the area of the activated surface. The method applied in Sections 5.1, 6.4, and 7.1 is based on determination of the activated area visualized on the HDPE plates using a 58 mN/m test ink. For plasma treatment, the plasma device is fixed with the PCPG tip positioned at a required distance $d$ from the substrate surface. The default distance and treatment time are 4.5 mm and 10 s, respectively. The method details are described in [11].

### 3.4. Thermal Characterization

It is known that, with increasing temperature, the input impedance and the resonance frequency of the RPT decrease [79]. It means that, for the constant power, the input current increases and the input voltage decreases with temperature. At the same time, the voltage transformation ratio also decreases [80]. The lowering of the PCPG output voltage results in less efficient discharge, and, consequently, in reduced production of chemically active species. On the other hand, it is known that the thermal stabilization of the RPT takes several minutes [81], causing variations of the device efficiency. It is also observed in PCPG that the amplitude of the output voltage and the number of micro-discharges per cycle of piezoelectric oscillation decrease with time of operation of non-cooled PCPG operated with 5 W [68].

This high sensitivity on temperature results in a high interest in thermal characterization of the PCPG. The typical temperature sensors are used for temperature control at such positions of the PCPG, where their electric connections do not cause the disturbance of the electric field or the damping of the oscillation. More flexible is the application of an IR camera, showing not only the temporal variation of the temperature, but also changes in its

spatial distribution. It was used for thermal characterization of the CeraPlas™ F operated in the configuration with the gap of 2 mm between the tip and the one-mm-thick Al₂O₃ plate positioned on the grounded iron block (see Section 6.2.1). During the IR recording, the CeraPlas™ F was cooled only by convection, hanging on the electric connection wires. Figure 5a shows the time dependence of the temperature in the zone depicted with black rectangle in the IR picture. It is a hotspot positioned directly at the electric connections of the CeraPlas™ F device. It is hotter than the hotspot on the tip of the CeraPlas™ F, where plasma is generated.

To support the statement that the temperature of the PCPG has a strong influence on its performance, Figure 5 shows the influence of the compressed dried air (CDA) flow on the oscillation voltage of the CeraPlas™ HF and on the number of micro-discharges per oscillation cycle. With decreasing gas flow, the cooling of the PCPG is getting insufficient, its temperature is rising, and it causes the drop of both performance values. This tendency correlates with a decrease of the ozone production rate with decreasing CDA flow, discussed in Section 4.4. The lack of sufficient cooling can cause not only the deterioration of performance, but also the fatal failure of the device, occurring in the case of ongoing thermal overload by dissolving of the soldered connections, thermal break, or local depolarization.

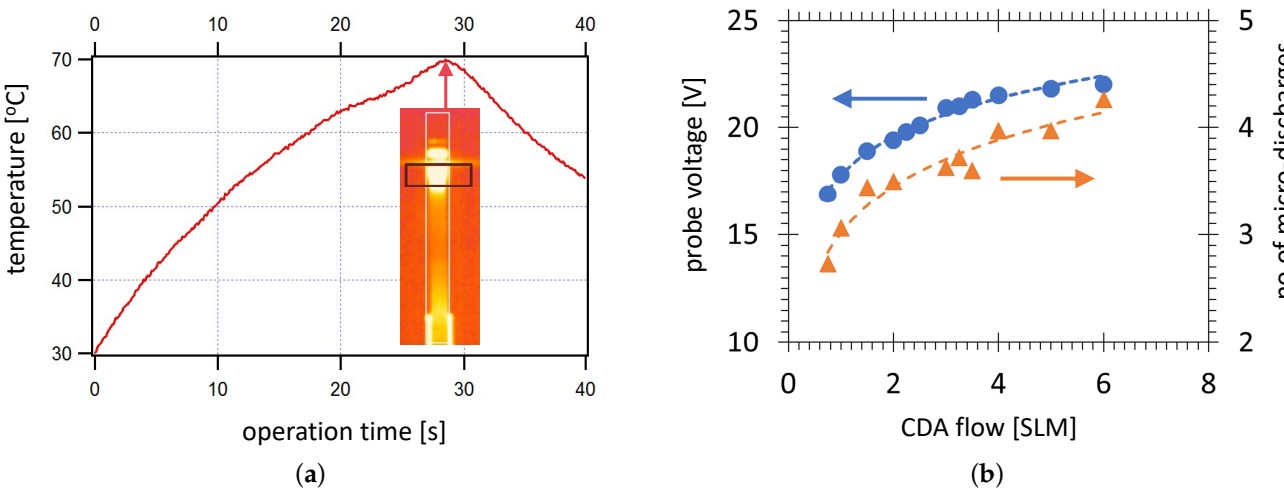

(**a**)  (**b**)

**Figure 5.** Thermal characterization of the PCPG. (**a**) the IR camera picture of the CeraPlas™ F with temperature trace collected at position depicted by rectangle; (**b**) influence of the CDA flow on the voltage and the number of micro-discharges per the oscillation cycle of CeraPlas™ HF.

The IR recordings help the plasma instrument designer to avoid such threats by proper dimensioning of cooling and choice of suitable materials for the PCPG holder. The thermal management measures such as pulsed operation mode [82] can also be investigated.

### 3.5. OES

Different spectral techniques can be used for characterization of APP plasmas. Due to its simplicity, the optical emission spectroscopy (OES) is frequently applied. In [83], the OES is used for characterization of a PDD ignited in a gas mixture of 95% He with 5% N₂ admixture. The excitation emission of N₂(C-B, 0-0) at 337 nm was used to determine the electron concentration in the PDD. The electric field to electron concentration ratio (E/N) in the corner of the piezo element is about 50 Td and electron concentration is $9 \times 10^{15}$ m⁻³ while E/N in the center of the element is about 40 Td and electron concentration is $3.1 \times 10^{14}$ m⁻³. In addition, the electron velocity distribution function is determined by comparing the ratio of excitation emission of N₂(C-B, 0-0) to that of N₂⁺ (B-X, 0-0) with the measured emission intensities.

The OES was also used for characterization of the piezobrush® PZ2 operated directly, with multi-gas nozzle (MGN), and near-field nozzle (NFN) [84]. All three spectra collected for air plasma are similar in shape, but differ in its intensity. The main emissions originate from the second positive system of $N_2$ and are located in the UV and visible spectral range. Overall, these are typical spectra for atmospheric pressure plasmas ignited in ambient air, e.g., like in [85].

## 4. PCPG for Ozone Generation

The established methods of ozone generation are by photochemical generation [86], for example by $Xe_2$ excimer lamps [87], or in electrical discharges [88,89]. The last can be subdivided in ozone generation by dielectric barrier discharge (DBD) in oxygen [46] and air [90], by surface dielectric barrier discharge (SDBD) [91,92] or by corona discharges in air [93], oxygen, and carbon dioxide [94]. However, the RPT driven discharges are also used for production of ozone [29,30], both in air and in oxygen [95,96]. The PCPG is generating ozone in different oxygen containing environments. In the next sections, the two modes of oxygen generation—in the gas flow and in the closed volume—are discussed.

### 4.1. Ozone Generation in Gas Flow

The generic setup used for the ozone generation in the flow of pure gas mixtures is shown in Figure 6a.

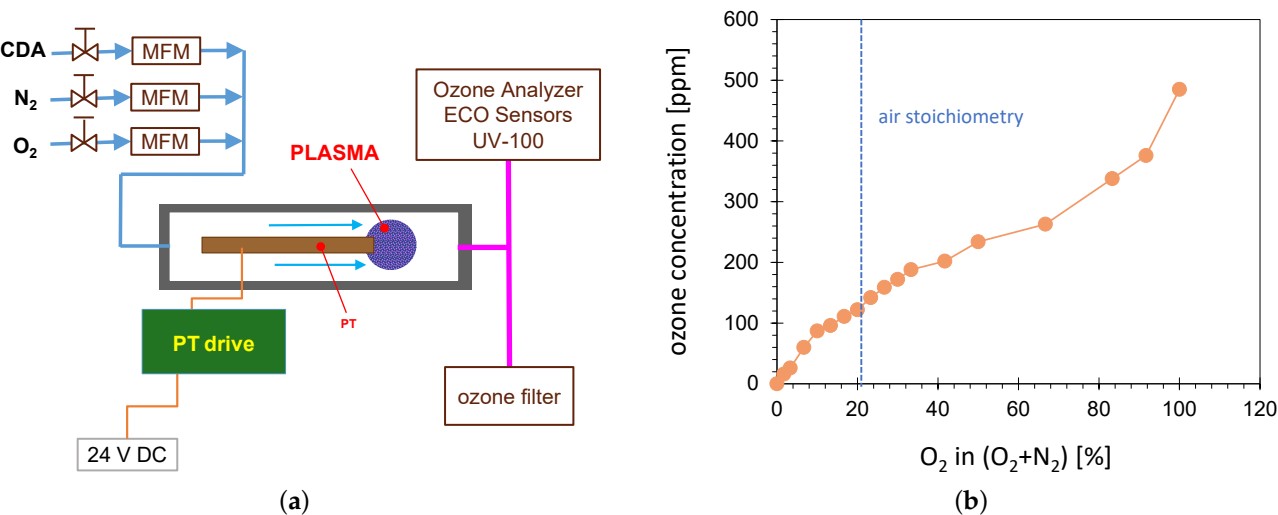

**Figure 6.** Measurement of the concentration of ozone generated by PCPG in the flow of pure gas mixtures. (**a**) setup; (**b**) ozone concentration as a function of oxygen percentage in an oxygen–nitrogen gas mixture, measured at a total gas flow of 6 SLM and PCPG power of 8.3 W.

To obtain exact ozone concentration values, the measurements have been conducted with CeraPlas™ F, embedded without driver electronics in the module with gas flow. The PCPG can be operated in CDA or in a mixture of nitrogen and oxygen. The CDA flow is controlled by a needle valve and MFM of FESTO. The nitrogen and oxygen flow are controlled by the use of MFC of MKS.

### 4.2. Ozone Concentration vs. Oxygen Percentage

Figure 6b shows the increase in ozone concentration with the increasing percentage of oxygen in the nitrogen–oxygen gas mixture. As is to be expected, the ozone concentration for pure nitrogen is measured as equal to zero. The maximum value of 485 ppm is achieved for pure oxygen. This value is higher by a factor of four than for 21% of oxygen in a nitrogen–oxygen gas mixture, which is close to the air stoichiometry.

### 4.3. Characterization of Ozone Production

The ozone concentration can be measured directly but is not a suitable value for characterizing RPT efficiency because the result strongly depends on air flow. A more suitable value for this purpose is the production rate, defined as mass of ozone produced per time unit.

Knowing the gas flow $f_{gas}$, the production rate of ozone $R_{prod}$ can be calculated from the ozone concentration $N_{O_3}$ using the following equation:

$$R_{prod} = \frac{M_{O_3}}{V_A} \cdot f_{gas} \cdot N_{O_3}, \tag{1}$$

where $V_A$ is the molar volume and $M_{O_3}$—the molar mass of ozone (48 g/mol).

### 4.4. Ozone Concentration vs. CDA Flow

The concentration of ozone as a function of CDA flow is visualized as a blue plot in Figure 7a. According to the fitting curve of this plot, the ozone concentration decreases almost inversely proportional to the CDA flow. The higher the flow, the stronger the dilution grade of ozone, and the lower the expected activation efficiency. To maximize the activation efficiency, the ozone concentration must be maximized by minimizing the air flow. The limiting factor regarding minimization of the air flow is the PCPT cooling requirements, which are dependent on the power coupled in the system.

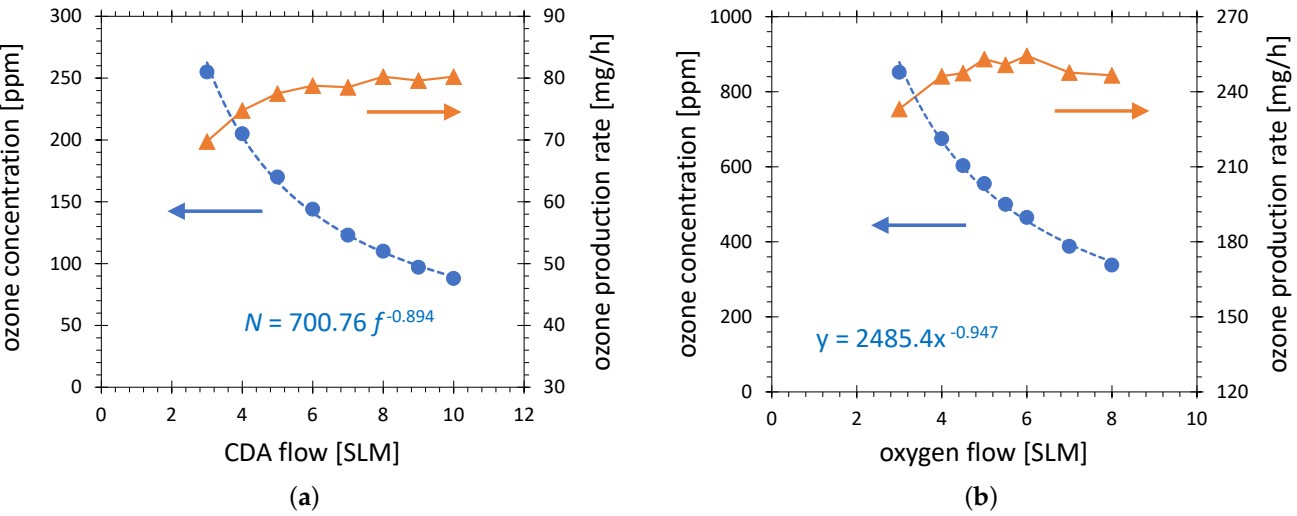

**Figure 7.** Concentration and production rate of ozone as a function of gas flow for (**a**) CDA (results from [11]) and (**b**) oxygen. The PCPG power is 8.3 W.

Using the ozone concentrations from the blue plot and Equation (1), the production rates are calculated and visualized as a red plot in Figure 7. For gas flow higher than 5 SLM, only a small variation in the production rate is observed. With gas flow decreasing below 5 SLM, the ozone production rate decreases. This effect can be explained by the increasing temperature of the PCPG due to insufficient air cooling, as already discussed in Section 3.4. The CDA flow below 3 SLM was not investigated, to avoid the thermal overload of the PCPG.

### 4.5. Ozone Concentration vs. Oxygen Flow

The disadvantage of the oxygen–nitrogen gas mixture for ozone generation is the production of nitrous oxide, dinitrogen pentoxide and other nitric oxides [97]. This drawback can be avoided by using pure oxygen.

The values of ozone concentration displayed are always collected 60 s after switching the plasma on. For shorter times, the value shown by the ozone gauge is not yet stable. For longer times, the ozone values start to decrease slightly, which can be related to the

increase in PCPG temperature. The maximum value of 852 ppm has been achieved for the oxygen flow of 3 SLM. The ozone concentration increases with decreasing oxygen flow. Furthermore, for oxygen, the PCPG operation was not investigated at flows below 3 SLM, so as to avoid thermal overload of the PCPG.

The ozone production rate changes only slightly with oxygen flow, reaching the maximum value of 254 mg/h for 6 SLM and minimum value of 233 mg/h for 3 SLM. These values are about three times higher than the production rate in CDA or ambient air.

### 4.6. Influence of Power on the Ozone Production Rate

In Figure 8a, the production rate as a function of the input power of the CeraPlas™ F is displayed for a CDA flow of 8 SLM. The production rate increases in power in the entire power range investigated. It reaches the maximum value of 73 mg/h for 8 W. This trend follows that of the number of micro-discharges per cycle as a function of power, shown in the same diagram. Since the micro-discharges are responsible for generating chemically active species, a correlation with not only the ozone production rate but also with surface activation area can be expected. In Figure 8b, the activation area of piezobrush® PZ3 as a function of the power is compared with the performance of the piezobrush® PZ2. The main difference is the stronger fan of the piezobrush® PZ2, causing more dilution of the chemical species and consequently smaller activation area in the entire range of investigated power.

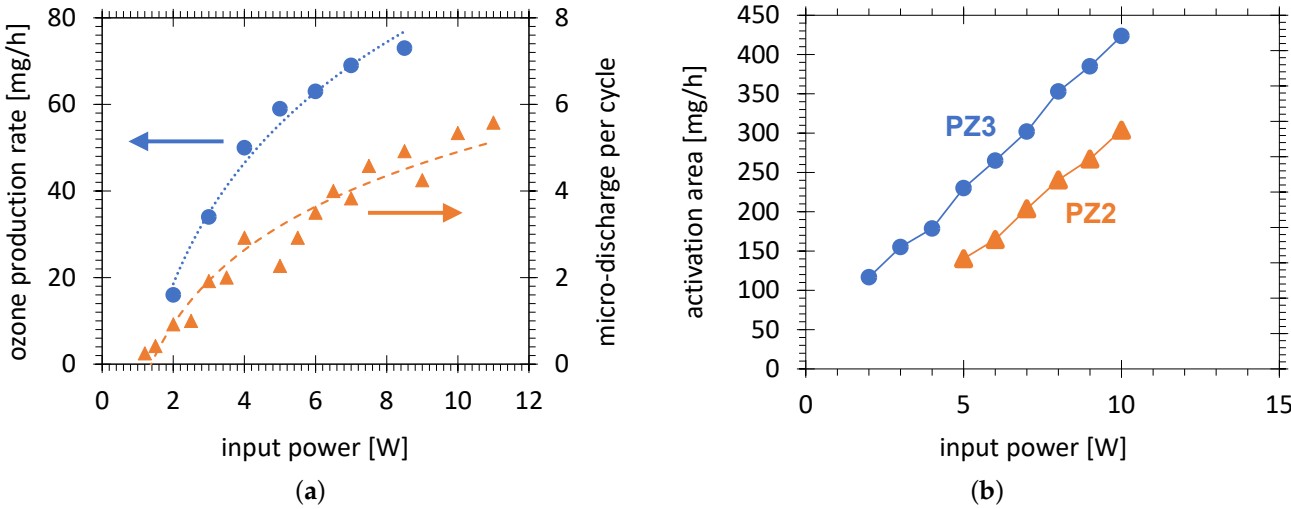

**Figure 8.** Performance of the piezobrush® PZ3 as a function of CeraPlas™ F input power. (**a**) ozone production rate and number of micro-discharges per oscillation cycle [11]; (**b**) activation area for piezobrush® PZ3 and for piezobrush® PZ2.

The increase in the ozone production rate is faster for power values below 5 W, and slows down for higher values, which can be interpreted as a loss of ozone production efficiency (production rate per energy unit). This effect can be explained by the increasing temperature of the PCPG tip. With increasing temperature, the following processes can affect the ozone production rate:

(i) Changes in the PCPG itself as already discussed in Section 3.4.
(ii) the less efficient ozone production. An increase of specific energy input in the discharge, due to higher power, leads to an increase of concentrations of nitrogen oxides, which react with atomic oxygen—the main species needed for ozone synthesis. This effect is known as discharge poisoning and can completely stop the ozone generation [98].
(iii) the enhanced decomposition of ozone. From 100 °C upwards, the thermolysis based on reactions with radicals [99] is an increasingly important loss mechanism of ozone.

### 4.7. Ozone Destruction

A number of factors causes the reduction of the half lifetime of ozone:

- high humidity contributing to ozone destruction due to chemical reactions with OH and HO$_2$ radicals [98],
- elevated temperature (see discussion in Section 4.6) causing the thermal decomposition [99],
- presence of surfaces containing water solutions with high pH-values [100,101],
- presence of carbon [102] and some organic substances,
- presence of catalytic materials such as metals and metallic oxides [103],
- UV illumination in the wavelength range causing photolysis [104], and
- high ozone concentration activating more efficient reaction channels for ozone destruction.

For the maximizing of the ozone concentration, such influences must be minimized. However, some of these mechanisms can be used for intentional destruction of ozone to avoid its release in the environment. The most widespread for ozone destruction is the application of the activated carbon filters. Their main drawback is that the activated carbon is consumed by reactions with ozone and have to be replaced after wear.

No significant wear is observed by catalytic filters, basing mainly on mixtures of MnO$_2$ and CuO$_2$ [105,106]. The problem of catalytic filters is that they are selective. It means that they remove ozone very efficiently, but are much less effective for some other, also harmful, substances present in plasma gases. These are either the nitrogen based oxidizing species or ozonides, resulting from chemical reactions of ozone with plastics. They can also be "poisoned" by a number of chemical compounds.

*4.8. Possible Ozone Applications*

The CeraPlas™ F based instruments can produce about 80 mg/h of ozone, and with oxygen, almost 250 mg/h. This allows for ozone concentrations of up to 1000 ppm in a flow control mode (see Section 4.5). Its efficacy was also proved in a bacteriological study with *Staphylococcus aureus* [107].

There are a large number of ozone applications which can be implemented on a small scale as a desktop or mobile system using one or more PCPGs.

- disinfection [108] and sterilization [109] for virus decontamination [110], for healthcare [109] or dental instruments preparation [111]
- bactericidal effects on *Escherichia coli* and *Staphylococcus aureus* [112]
- disinfection of food products, e.g., *Salmonella infantis* on the skin of chicken portions [113] or on eggs [114] or other microbial contamination on pork meat [115], post-harvest fresh fruit treatment [116]
- disinfection of goods inside a closed package [117]
- prolongation of the shelve time in the packaging e.g., on table grape berries [118]
- chemical contamination neutralization, for example, pesticide residue from the surface of vegetables [119,120], or fungicide residues on fresh produce [121],
- control of odor [122,123] and indoor bioaerosols [124].
- decomposition of volatile organic compounds (VOC) in air flow [125].
- bubbling ozone in water to decompose organic compounds [126,127]
- ozonated water for removal of pesticide residuals on vegetables [128]

## 5. PCPG Based APPJs

Thanks to its small weight and sizes, the PCPGs are very suitable for application in hand-held APPJs [129–131] such as piezobrush® PZ2 [11,132] or piezobrush® PZ3 [51].

*5.1. Activation Area*

The most important parameters having an influence on the activation area are: the treatment time, the PCPG input power, and the distance between the PCPG tip and the treated surface. The dependence on power is discussed in Section 4.6. In following, the influence of the treatment time and the distance are discussed.

### 5.1.1. Dependence on the Treatment Time

In the case of not moving PCPG, the plasma treatment duration can be controlled by switching on and off the power. In the case of movable PCPG, the speed and the size of the treatment zone are crucial for how long a specific substrate point is affected by plasma. Intuitively, we expect that the longer the treatment time, the larger the activated area. This expectation is confirmed by a monotonous increase of the activation area with treatment time displayed in Figure 9a. However, a clear saturation of the area values for longer times is observed. This trend can be explained by three mechanisms causing the limitation of the radial spreading of the activation zone: *(i)* increasing dilution of the chemically active species in the ambient air with increasing distance from the nozzle opening blowing the plasma gases, *(ii)* the decrease of the reactivity of the short-lived chemically active species with flow time, and *(iii)* the decrease of the CeraPlas™ F electric field with distance from its tip. The maximum size of the activation area typically doesn't exceed 26 mm, even by a very long treatment time.

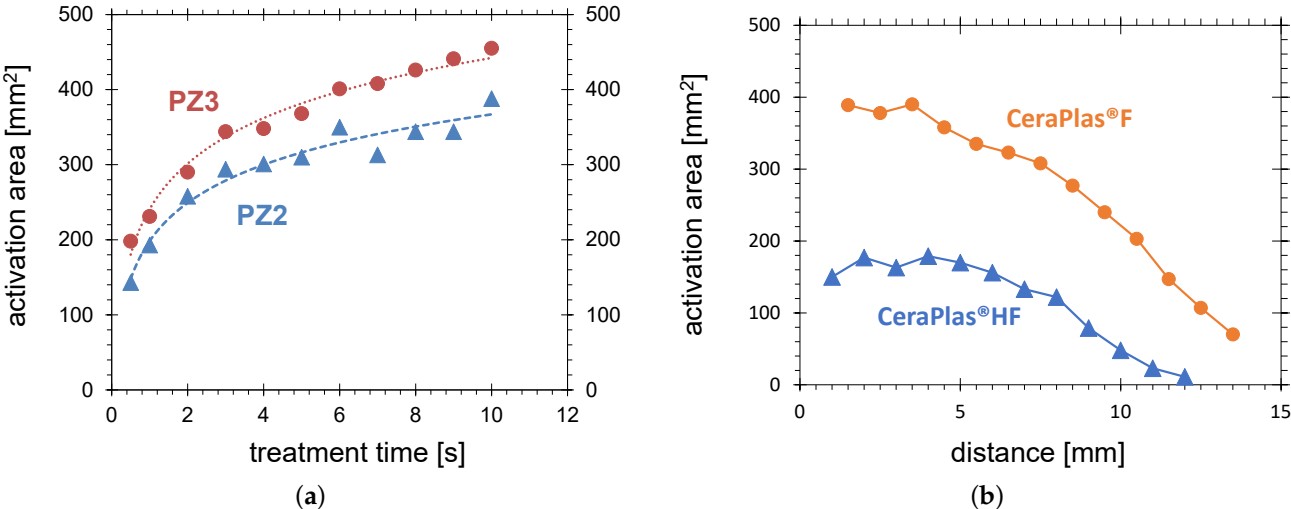

(**a**)  (**b**)

**Figure 9.** The activation area of the piezobrush® PZ3 operated with the power of 8.0 W. (**a**) the dependence on the treatment time for PCPG positioned in the distance of 4.5 mm from the HDPE substrate. For comparison, the results for piezobrush® PZ2 are included [51]; (**b**) the dependence on the distance for the treatment time of 10 s [11]. For comparison— the results obtained with 72 mN/m test ink on ABS substrates using the CeraPlas™ HF with 4.0 W power and no air flow positioned 4.5 mm from the substrate.

In Figure 9a, the time dependent activation results of the piezobrush® PZ2 are shown as blue triangles for comparison. The general shape of both plots is very similar, but the activation area for the piezobrush® PZ3 is typically 20–25% larger than for the piezobrush® PZ2, despite of the slightly higher power of 8.3 W compared with 8.0 W for piezobrush® PZ3. Similar as for curves in Figure 8b, larger activation area for piezobrush® PZ3 is explained by reduced air flow and consequently less dilution of the chemically active species.

### 5.1.2. Influence of the Substrate Distance

Of high practical importance is the information, in which distance from the treated surface of the piezobrush® PZ3 should be held, to achieve the optimum treatment result. To answer this question, the dependence of the activation area on the distance is investigated. The activation area decreases with distance increasing over 3.5 mm (see Figure 9b). This drop can be explained by the same three mechanisms mentioned in Section 5.1.1. The maximum of the activation area is reached quite close to the nozzle, 1.5 to 3.5 mm from the CeraPlas™ F tip. For a distance larger than 13 mm, the activation area is vanishing.

The second curve in Figure 9b represents the activation results obtained with CeraPlas™ HF. In comparison with the CeraPlas™ F used in the piezobrush® PZ3, the CeraPlas™ HF

is much smaller and, consequently, it is able to dissipate less power (compare the data in Table 1). This is the main reason for the much smaller maximum power of 4.0 W and smaller activation areas for CeraPlas™ HF, even though the determination of the activation area was performed with 72 mN/m test ink on ABS substrates, giving typically larger activation area. Common for both curves is their general shape. The best results are reached for the distance below 4 mm and a rapid decrease is observed for distance over 10 mm. Since the CeraPlas™ HF curve is obtained without gas flow, the importance of the electric field for the efficiency of surface activation process is highlighted.

*5.2. Application Examples*

A large number of applications of piezobrush® PZ2 and piezobrush® PZ3 for surface modification of different polymeric substrates is known. The considerably new disciplines of APP application are the plasma agriculture [133] and the plasma farming [134]. This is why in the next section the results for unconventional substrates—fruits—are shown.

5.2.1. Treatment of Fruits

The purpose of the fruits and vegetables plasma treatment can be:

- Disinfection and inactivation of spores such as *Penicillium digitatum* [135].
- Control of fungal plant pathogens [136] (e.g., *Aspergillus flavus* by surface barrier discharge [137]).
- Deactivation of spores.
- The decomposition of the pesticides on fruits and vegetables.
- In package decontamination (e.g., *E. coli* and *L. innocua*) of fresh strawberries and spinach [138].
- Seeds treatment for improving harvest [139].

In the study presented here, the limes, lemons, mangos and apples were treated by plasma generated by the CeraPlas™ F module operated with CDA or nitrogen. Table 2 contains the comparison of the DI water droplet contact angles measured on untreated and plasma treated surface of these fruits. All fruits with the exception of bio-apple were from the super-market. The bio-apple was directly from the garden, in which no spraying of chemicals is applied.

**Table 2.** Contact angle of a DI water droplet measured on the pristine and PDD treated fruits. Distance: 15 mm, CDA flow: 7 SLM, treatment time: 2 min.

| Fruit | Pristine | Treated |
|---|---|---|
| lime | 102° | 45° |
| lemon | 100° | 56° |
| mango | 102° | 60° |
| apple | 101° | 65° |
| organic apple | 103° | 60° |

5.2.2. Apple

Some limitations of plasma processing like increase in oxidation of lipids, reduction in color, decrease in firmness of fruits, and increase in acidity, etc. were reported [140]. The treatment of the fruits with PDD at small distances, less than 15 mm, poses a risk of the skin puncture. The fruits are consisting of electrically conducting electrolytes, causing the PDD to work in the spark mode, which can locally reach high current densities being able to damage the fruit surface. Therefore, for investigation of the dependence of the plasma treatment result as a function of distance between the fruit surface and the tip of the CeraPlas™ F, only distances ≥ 15 mm are considered. As the results in Figure 10a show, no significant reduction of the contact angle is observed for distance > 50 mm. Between 15 and 20 mm, a strong change of the contact angle is observed.

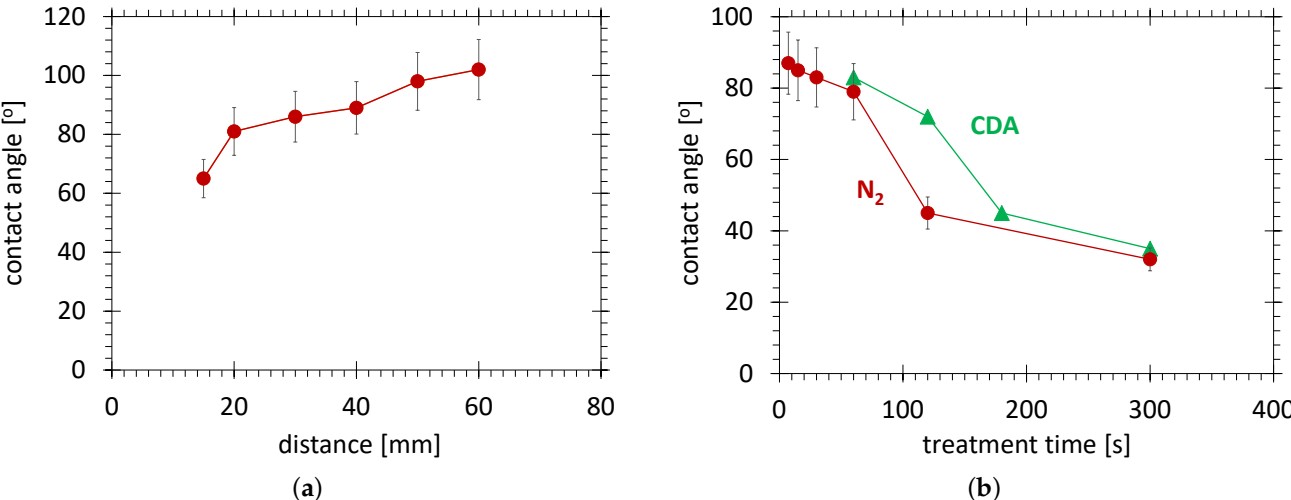

**Figure 10.** The contact angle of the DI water on the fruit surface treated by PDD (**a**) on the apple surface treated 2 min, and (**b**) on the lime surface at a distance 15 mm and 35 mm for CDA and nitrogen, respectively. The gas flow is 7 SLM for all experiments.

### 5.2.3. Lime

The contact angle measured with DI water on the untreated lime surface is 103°. It is decreasing strongly with increasing treatment time (see Figure 10b). For treatment time below 1 min in CDA plasma, only very marginal improvement of wettability is observed.

The contact angle is decreasing strongly with increasing treatment time in nitrogen remote plasma. For treatment time below 2 min, a moderate improvement of the wettability is observed. For very long treatment time (5 min), the contact angle as low as 35° is achieved.

## 6. PDD for DBD Generation

The PDD, applied directly to the electrically conducting substrates, causes sparks that can be damaging for thermally sensitive materials such as thin aluminum foil, carbon fiber reinforced plastics (CFRP), micromechanical and microelectronic structures or organic tissues. The power density of the PDD can be reduced applying a dielectric barrier between the PCPG and the substrate. In the next sections, the application potential of PDD driven DBDs, their configurations, performance evaluation, and application examples for surface treatment are presented.

### 6.1. DBD Application Potential

The dielectric barrier discharge (DBD) is the oldest [141,142] and very versatile [143] low temperature atmospheric pressure plasma (LT-APP). The size of DBD generators ranges from micrometers by micro-discharges used for ignition of atmospheric pressure glow (APG) [1] or in plasma panel displays (PPD) [144], to large volume (m$^3$-range) systems used for industrial scale production of ozone [46] chemical processing [145], or air purification [146–150]. The DBDs differ not only in size but also in the excitation scheme. The main differentiation is between the volume DBD ignited in gas gaps between the dielectric barrier and the electrode, and the planar DBD, spreading over the dielectric barrier surface [151]. To the first category belong the single gas gap DBD with one or two dielectric barriers and the double gas gap DBD with micro-discharges ignited on both sides of the dielectric barrier. To the second category belong the surface DBD (SDBD) [56,91] and the coplanar surface DBD (CSDBD) [152].

Such DBD generators are a focus of our interest, which can be powered by PCPG. It means that the operating power should range from 2 to 10 W. Even though the DBD is used also for treatment of electrically non-conducting substrates placed on electrically conducting background [153,154], the focus here is on conducting materials, because the non-conducting ones can be treated more efficiently using the PDD without any dielectric barrier, as described in Section 5. The example applications that can be implemented using PCPG driven DBD are the pretreatment of the aluminum alloy surface for enhanced adhesion of a polyurethane coating [155], hydrocarbon contamination removal from the flat aluminum surface [156], removal of lubricant residua from silicon or steel surfaces [157], or cleaning of metal wires [158,159]. The treatment of partially electrically conducting organic materials, like wood for improvement of paintability [160,161], cotton fabrics for control of resizing and wettability [162] and for smell reduction [163], or seeds for improvement of germination [164,165] can be also implemented.

PCPG driven DBDs have a high application potential for disinfection and sterilization of e.g., surfaces of fabrics [166] and nonwoven fabrics [167], or catheters [168]. In [169], the reduction of *bacillus subtilis* and *aspergilius niger* spores is demonstrated. The inactivation of biological components of bacteria and bacteriophages is investigated in [170].

A broad class of DBD applications represent the coating processes such as deposition of polymer films [171–173], especially such using the vapor monomer gas mixtures in noble gases [174–178]. However, they are not a focus of this review.

The physical principles of the DBD are the subject of numerous experimental studies [179] and simulation works. The DBD discharge mechanisms and properties are described in several reviews [143,180] and will not be repeated here.

### 6.2. Configurations of PDD Driven DBD

6.2.1. PCPG as Electrode of Single DBD

The simplest discharge architecture for igniting the DBD by use of a PDD is shown in Figure 11a. The PDD ignites between the PCPG and the surface of the dielectric barrier, behind which the grounded electrode is placed [64]. This generic scheme is known in several variants. The PZT RPT was used for ignition of a DBD between the surface of the RPT and the ITO-electrode deposited on the backside of the dielectric barrier. The microdischarges produced in the narrow gap between the high-voltage side of the RPT, and the dielectric barrier was observed optically through the transparent ITO-electrode [181]. Similar discharge configurations are used for ozone production [95]. The up-scaling of the ozone production rate was achieved by a parallel connection of six RPT devices [29]. In the version of the device equipped with Suprasil quartz barrier and mash electrode and operated with argon or xenon, the excimer VUV radiation is produced [182]. This discharge configuration corresponds to the one used for surface activation of polymer samples, which play the role of the dielectric barrier, described in Section 3.3.

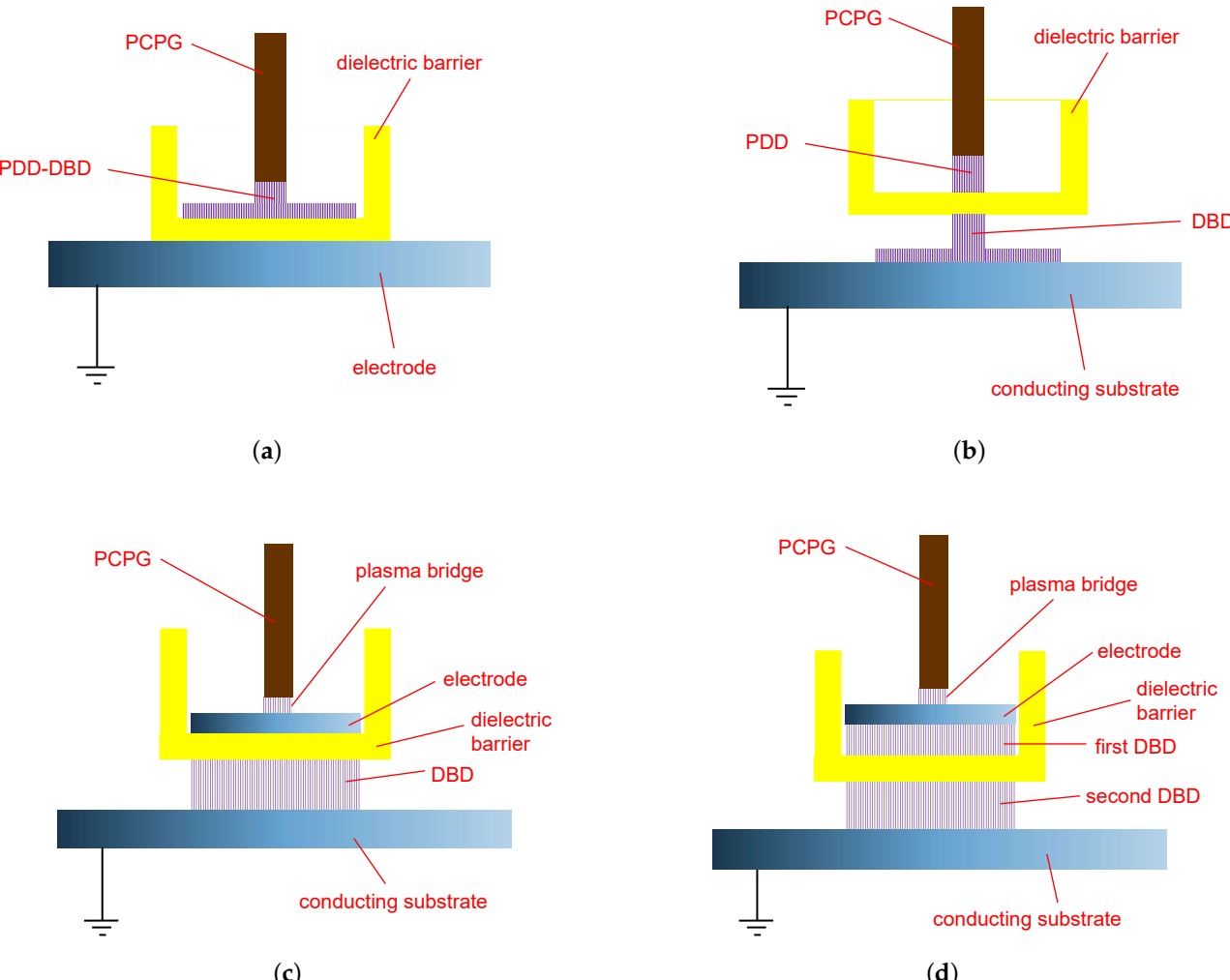

**Figure 11.** Four configurations of the PDD driven DBD. (**a**) the PCPG as the DBD electrode in a single DBD, (**b**) the PCPG as the DBD electrode in a double DBD, (**c**) the excitation electrode powered by PCPG over the plasma bridge and the single DBD, and (**d**) the excitation electrode powered by PCPG over the plasma bridge and the double DBD.

6.2.2. PDD as DBD Electrode of Double DBD

A small modification of the discharge configuration shown in Figure 11a, making a gap between the electrode and the dielectric barrier, results in the configuration shown in Figure 11b, being used as a so-called near-field nozzle (NFN) of the piezobrush® PZ3 for surface treatment of electrically conducting materials. In this configuration, the energy from the PCPG is transferred to the substrate in three steps:

(1) from PCPG high voltage surface to the surface of the dielectric barrier by PDD,
(2) capacitively coupled across the dielectric barrier, and
(3) from the dielectric barrier to the substrate surface by DBD micro-discharges.

The drawback of this configuration is that the pattern of the activated zone is strongly affected by the shape of the PDD plasma, which follows the rectangular shape of the high voltage face of the PCPG. The further drawback is high sensitivity of the activation area on the distance between the substrate and the dielectric barrier. To obtain the circular activation area, the DBD could be generated by a circular electrode positioned on the PCPG facing side of the dielectric barrier. The problem is the electric connection between the tip of the PCPG and such electrode. Any electric connection on the tip of the PCPG results in

damping of the PCPG resonant oscillation and deterioration of the device performance. Such connections are also the lifetime limiting factor for the device.

### 6.2.3. PDD Plasma Bridge

Any mechanical connection between PCPG and the electrode can be completely avoided, if the electric contact is created by a gaseous discharge. Since the purpose of such a discharge is not the plasma process itself, but creating the galvanic connection between the main plasma and the electrode, the term plasma bridge established for low pressure plasmas [183,184] will be used.

The application of the plasma bridge allows to avoid the mechanical damping of the PCPG oscillation. Consequently, a high resonant quality factor of the PCPG oscillation can be achieved. The disadvantages of such a solution are ohmic power losses in the plasma bridge and heating up of the PCPG tip. Despite of these drawbacks, all extension nozzles for the commercial plasma generator piezobrush® PZ2 [185] are based on such a power coupling principle.

If the metal electrode is positioned sufficiently close to the PCPG high voltage surface, the spark is established between them. For a short time, the electric charge accumulated on the PCPG surface can flow to the electrode, causing a change of its potential. Consequently, the potential of the electrode is following the voltage signal of the PCPG output. Opposite to the configuration without a metal electrode, the amount of the electric charge transferred through the micro-discharge is not limited by the surface capacity of the dielectric barrier, but by the much higher output capacity of the PCPG, resulting in higher efficiency of the DBD discharge.

### 6.2.4. DBD Electrode Biased over the Plasma Bridge

The configuration with electrode powered over PDD plasma bridge is shown in Figure 11c. This construction corresponds to the well-known floating electrode DBD (FE-DBD) [186] with the difference that the active electrode is not biased from HV cable but from the PCPG over the plasma bridge. The prototype of a FE-DBD nozzle is developed for use with piezobrush® PZ3.

The characteristic property of the FE-DBD is that the second electrode of the DBD discharge is the treated object (e.g., biological substrate, tissue, body of a living being) [186]. The applications are focused on physical and biological mechanisms of direct plasma interaction with living tissue [187]. The toxicity of the cold plasma treatment for the wound on live pig skin tissue is one example [188]. Applications in dentistry and oncology are also known [189]. Living tissue sterilization [190] including open wounds (live rat model) [191] and sterilization of *Escherichia coli* [192] are claimed to have been successful.

The proximity nozzle of the piezobrush® PZ2, shown in Figure 12a, is also based on this principle. Its electrode consists of two parts: the stainless-steel coupling electrode and the electrically conductive glue filling the gap between the coupling electrode and the surface of the dielectric barrier. The aim of this construction is to distribute the DBD power over large area, to reach the power density low enough for treatment of living tissue.

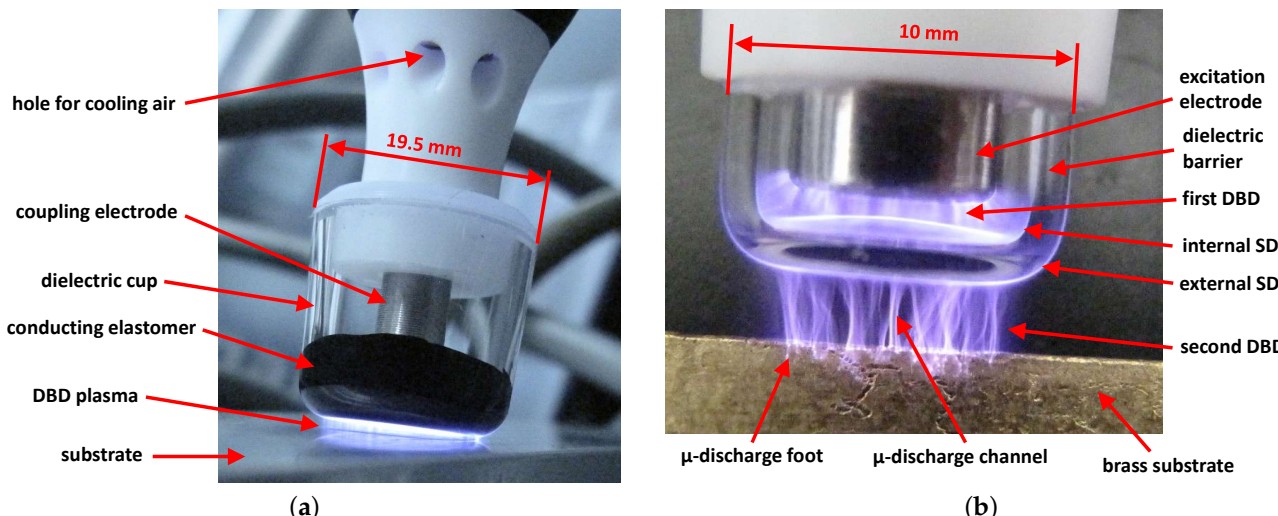

**Figure 12.** Two versions of the DBD producing, plasma bridge powered nozzles of the piezobrush® PZ2 with explanations. (**a**) the proximity nozzle with single DBD; (**b**) the near-field nozzle with double DBD.

### 6.2.5. Double DBD Driven over the Plasma Bridge

The production of a durable and robust electrode without letting a micro-air-gap between it and the dielectric barrier is challenging. To overcome this technological problem, for the near-field nozzle of the piezobrush® PZ2, the solution shown in Figure 11d is implemented. The DBD electrode powered over the plasma bridge from the PCPG is generating a double DBD, without touching the dielectric barrier. In this construction, the power flow from the PCPG to the substrate follows four transfer steps:

(1)    from the HV surface of the CeraPlas™ F to the excitation electrode,
(2)    from the excitation electrode to the internal surface of the dielectric barrier,
(3)    across the dielectric barrier, and
(4)    from the outer surface of the dielectric barrier to the substrate surface.

Figure 12b shows the double DBD discharge, present during the treatment of a metallic substrate by use of the near-field nozzle of the piezobrush® PZ2. The surface discharges (SD) visible on the inner and outer surface of the dielectric barrier contribute to the widening of the discharge.

### 6.2.6. SDB Driven by PDD

If the configuration shown in Figure 11a would be modified by exchanging the solid electrode for a mash, the discharge configuration analogue to the well-known surface micro-discharges (SMD) [193] could be created. The only difference is the biasing of the discharge electrode not directly from the HV cable but from the PDD.

The serious drawback of the volume DBD by treatment of large area dielectric substrates is that the discharge channels have to cross the substrate material and can cause its local overheating or even damage [194]. This is not the case for different types of planar DBDs because the discharge channels run parallel to the treated surface.

The surface micro-discharge (SMD) is a variant of the surface dielectric barrier discharges (SDBDs) [92,195]. At a low power level, it can be used for the treatment of biomolecular films [196]. The SMD was applied in plasma treatment of onychomycosis [197], in cancer research [198–201], in in-vivo skin treatment [202], and in the preventive medicine for nosocomial infections [203,204]. The links between antimicrobial effects and plasma chemistry were studied [205], and the efficacy of plasma in regard to spores [206] and bacterial decontamination (*Escherichia coli*) were investigated [207].

*6.3. PDD Driven DBD Characterization*

6.3.1. Discharge Focusing

With decreasing distance between the dielectric barrier and the substrate surface, the number of discharge channels per surface area increases. The impression is that the discharge gets more homogeneous. It can be expected that, under such conditions, the surface treatment will be also more homogeneous as shown in Figure 12b.

If the distance is getting too large, the discharge channels between the barrier and the substrate disappear. The electric field in this zone is getting too weak for ignition of plasma. Figure 13a shows that the discharge between the excitation electrode and the internal surface of the dielectric barrier remains sustained, but it has no effect on the substrate surface in the sense of activation.

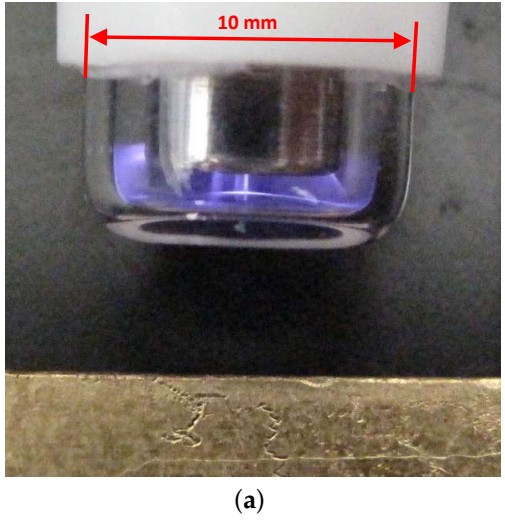 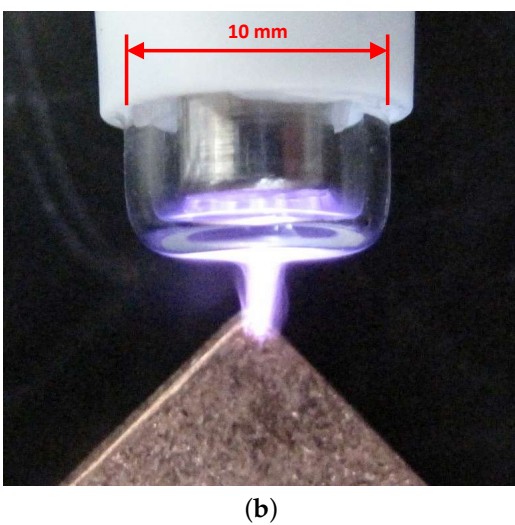

| (**a**) | (**b**) |

**Figure 13.** Influence of the distance and substrate shape on the structure of the discharge. (**a**) Too large distance—second DBD disappears, and (**b**) focusing of the micro-discharges at one spot on the sharp edge of the substrate.

The sharp edges on the surface of an electrically conducting substrate cause the local increase of the electric field, which is promoting the ignition of a discharge channel just on such an edge (see Figure 13b). Such promoted plasma ignition can be obtained not only on the front face of the dielectric barrier, but also through the cylindrical part of it.

6.3.2. DBD Current Pulses

To the electrically conducting substrate positioned close to the dielectric barrier in the DBD discharge configuration shown in Figure 11b, short current pulses are flowing. They can be measured by a current probe in the wire connection between the substrate and the ground, as shown in the inlay drawing in Figure 14a. Each such pulse represents a single DBD micro-discharge.

The DBD micro-discharge with a duration of a few ns excites a damped oscillation in the measurement circuit. This electric response depends on the length of the grounding wire, size of the grounded plate, and the surroundings of the measurement system. It is why it is difficult to make any conclusions about the DBD discharge on the base of the amplitude, dumping constant, or frequency of these oscillations. The value, which is independent on such influencing factors, is the number of micro-discharges. By counting the electric responses, the evaluation of the DBD is possible. The number of such pulses flowing per single cycle of the PCPG oscillation, can be used for evaluation of the discharge intensity.

Figure 14a shows the dependence of the pulse number per cycle on the gap between the PCPG and the dielectric barrier. With the decreasing gap, the number of pulses per cycle increases. It is a known effect in DBD gaps and can be explained by increasing electric field. This increase stops for gaps smaller than 0.2 mm. The most probable reason is the

weaker cooling of the PCPG tip and, consequently, increase of its temperature, which in turn results in the flattened resonance curve of its oscillation, lower voltage transformation ratio, and, consequently, a lower electric field in the gap.

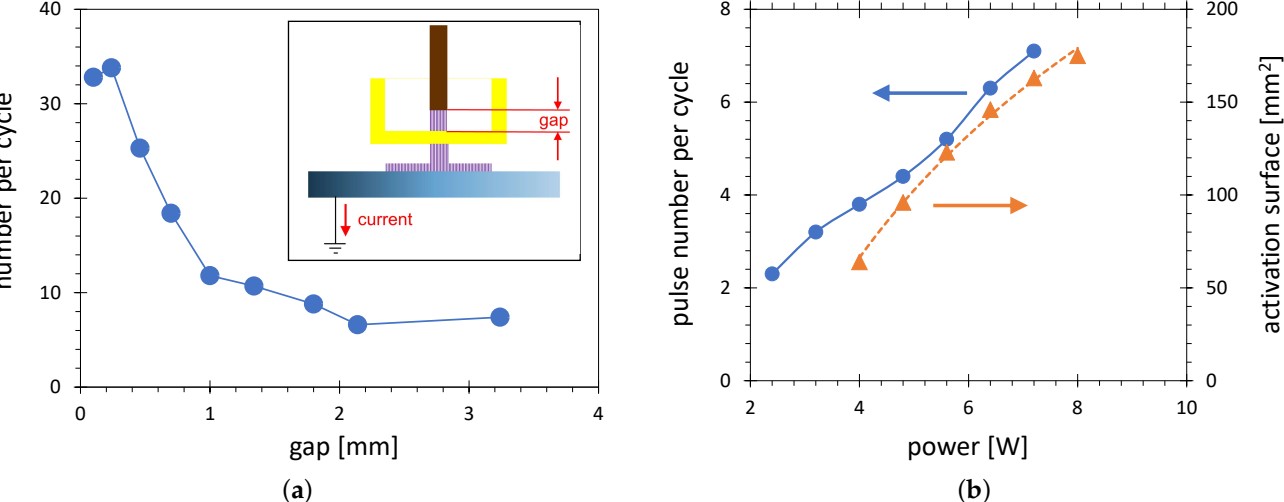

**Figure 14.** The number of the current pulses flowing per one cycle of PCPG oscillation between the dielectric barrier and the ground as a function of (**a**) the gap between the PCPG tip and the dielectric barrier (see the inlay drawing), and (**b**) the PCPG power for the distance between the substrate and the barrier of 0.5 mm, with dependence of the activation area on the PCPG power included for comparison. Each point represents the mean value out of 12 cycles of PCPG oscillation. The current probe TCP202, Tektronix, Inc.: Beaverton, Oregon, United States, is used.

### 6.4. Surface Activation Area

6.4.1. Influence of Power

Figure 14b shows the activation area as a function of PCPG power for the piezobrush® PZ3 near-field-nozzle (electrode configuration as shown in Figure 11b). Similar to the open PDD (see Figure 8), it correlates with the curve showing the dependence of the number of micro-discharges per cycle of the PCPG oscillation on power.

6.4.2. Comparison of Different DBD Configurations

For comparison of different PDD powered DBD configurations, the activation area is used again. Figure 15a shows two curves representing the activation area as a function of the distance between the substrate and the outer surface of the dielectric barrier (quartz cup). The activation area for configuration shown in Figure 11c is twice as large as the values for configuration shown in Figure 11b. The explanation of this difference is the much more efficient energy transfer over the plasma bridge than over the PDD-DBD.

Figure 15b demonstrates The performance difference between the configurations shown in Figure 11c (double DBD: PZ3) and Figure 11d (single DBD: PZ2), respectively. Since the activation efficiency of the NFN of the piezobrush® PZ2 on HDPE is not sufficient for evaluation of the activation area, the less demanding ABS substrates are used. The two curves show the dependence of the activation area on the surface energy determined by test inks. As expected, the activation area is decreasing with the surface energy visualized for both configurations. The activation areas for single DBD is in the entire range of surface energy almost twice as large as for the double DBD. The lower efficiency of the double DBD is the result of more energy transfer stages and, consequently, more power losses on the way from the PCPG to the substrate.

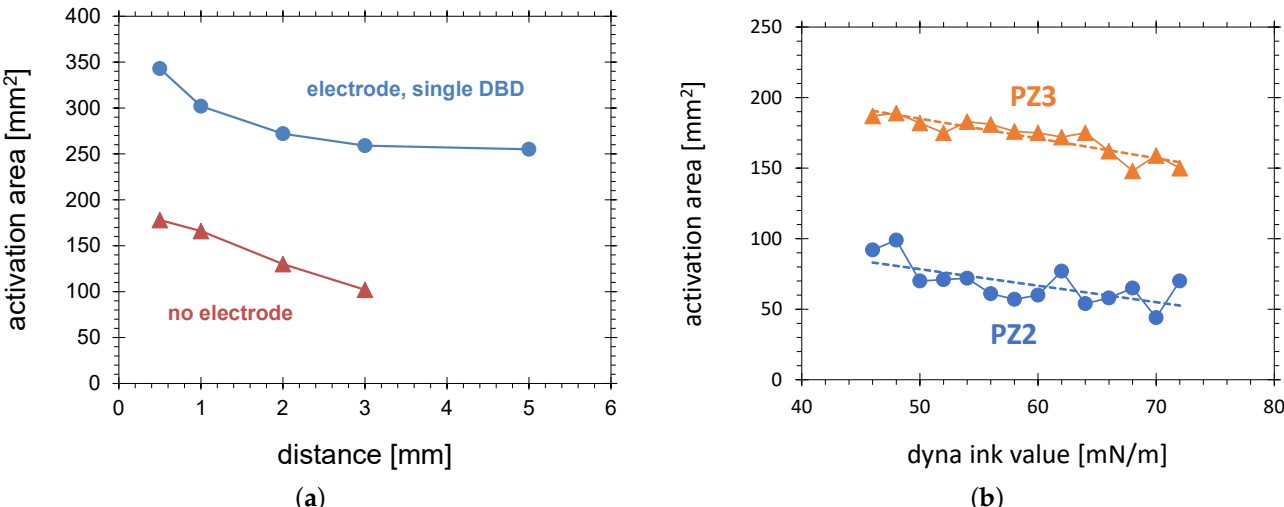

**Figure 15.** Comparison of activation area for different DBD configurations. (**a**) dependence of the activation area on HDPE visualized using test ink gauged 58 mN/m on the distance between the dielectric barrier and the substrate for the DBD configuration without excitation electrode (see Figure 11b) and with excitation electrode and single DBD (see Figure 11c); (**b**) dependence of the activation area visualized on the ABS substrate with test inks gauged from 46 to 72 mN/m for single (PZ3) and double (PZ2) DBD. Treatment time: 10 s. PCPG power: 8 W. Distance between substrate and the dielectric barrier: 0.5 mm.

### 6.5. Example: Treatment of Titanium

Titanium is one of the major materials used in implantology. Different types of plasma are used successfully to improve the biocompatibility of the titanium surface. For example, the APPJ is used to improve the behavior of oral soft tissue cells [208]. In addition, the piezobrush® PZ2 is used for such purposes [209,210].

In this review, the application of the PCPG driven DBD configuration with single DBD for improvement of titanium wettability is investigated. In Figure 16a, the dependence of the contact angle on the treatment time is demonstrated. A rapid decrease of the contact angle (increase of the surface energy) is observed within the first second of the treatment. Longer treatment causes only a slow reduction of the contact angle. The minimum contact angle reached after 20 s treatment is 17°.

The moderate hydrophobic recovery of the titanium surface can be observed. In Figure 16b, the decrease of the activation surface as a function of the storage time is shown. For visualization of the activation, the test ink gauged for 72 mJ/m$^2$ was used. It can be observed that the hydrophobic recovery is quite slow during the first 24 h after treatment. After the next 24 h, the activation area is reducing on more than 30%. The hydrophobic recovery can be accelerated by thermal treatment. After 60 s heating on oven at a temperature of 380 °C, the activation area visualized by 72 mJ/m$^2$ ink is vanishing.

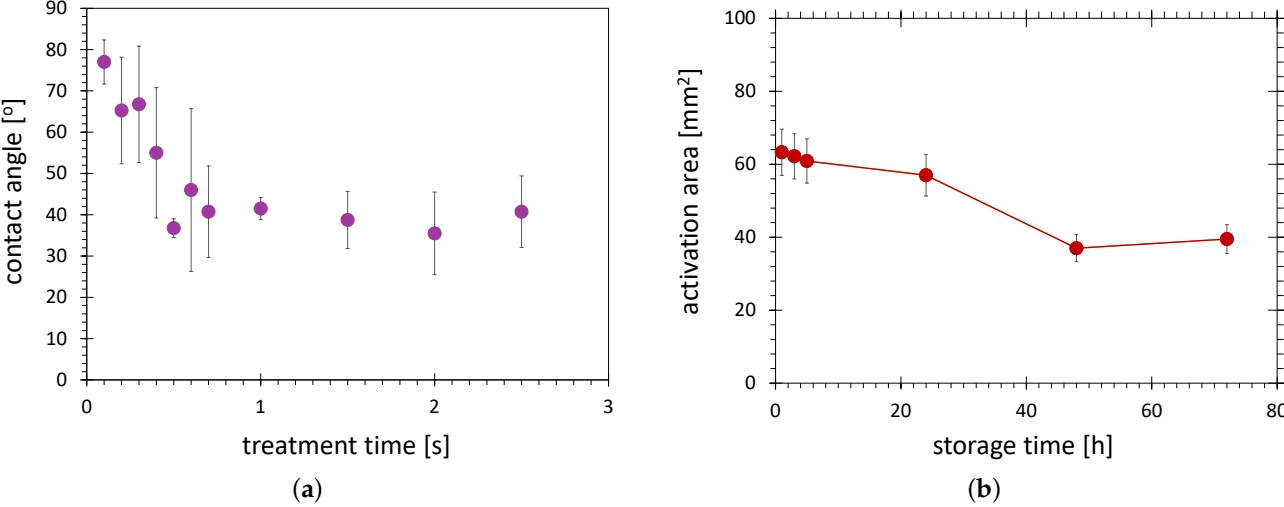

**Figure 16.** (**a**) the contact angle of water on titanium surface as a function of the treatment time and (**b**) the activation area visualized by use of 72 mJ/m$^2$ test ink on a titanium surface as a function of storage time after 10 s treatment. The plasma treatment is conducted with piezobrush$^®$ PZ3 with a near-field module at a distance of 1 mm from the titanium plate. The treatment and storage were in ambient air (temperature of 20 °C, relative humidity of 50%).

### 6.6. Example: Soot Removal

A broad class of DBD applications is the removal (cleaning) of the organic films/residua from the electrically conducting surfaces described in Section 6.1. As an implementation example for the PDD driven DBD removal process, Figure 17a shows the statical removal of a soot film from the stainless steel surface. The pattern eroded in the soot film is shown in Figure 17b and demonstrates the micro-discharge structure of the DBD ignited using the piezobrush$^®$ PZ2 equipped with near-field nozzle (NFN). At places where the micro-discharge is impacting again and again, blank stainless-steel spots are eroded, as shown in Figure 17c. The eroded spots are placed among the darker areas, where soot was not removed completely. To achieve complete removal, either the increase of the treatment time or the movement of the plasma generator must be applied.

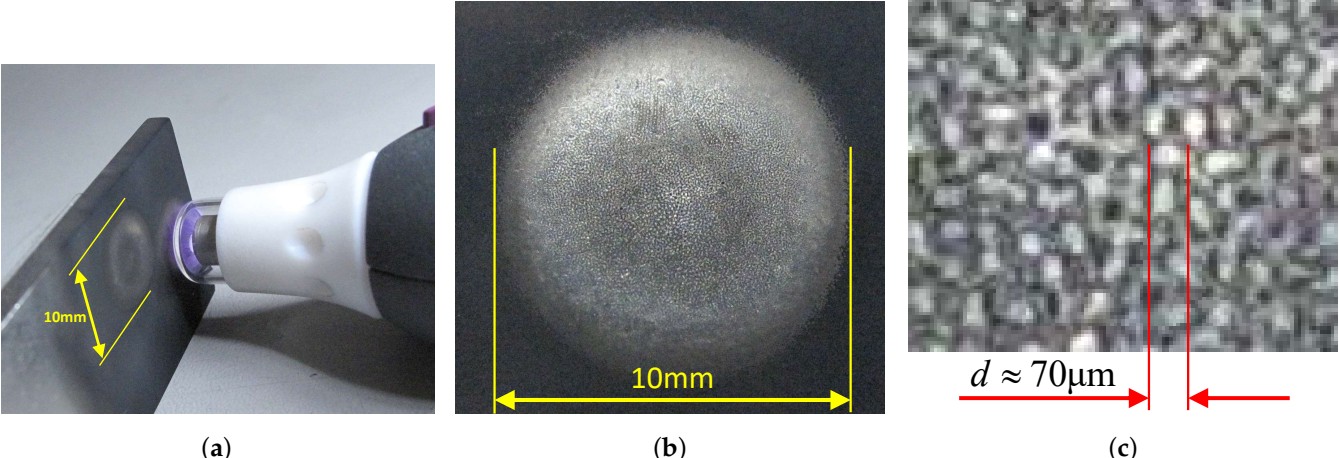

**Figure 17.** Soot removal from the surface of a stainless-steel substrate using the piezobrush$^®$ PZ2 with the near-field nozzle (NFN). (**a**) the picture of the DBD discharge during treatment; (**b**) the pattern eroded in the soot film during the 30 s treatment; (**c**) the magnification of the eroded soot film with depicted size of a single erosion spot.

## 7. PCPG Powered Needle Discharge

*7.1. Operation Principle*

The main component of the needle nozzle is the needle electrode (see Figure 18a). Similar to the DBD electrode, the needle electrode is biased over the PDD plasma bridge (see Section 6.2.3) short-cutting electrically the PCPG and the needle electrode head. To avoid the thermal damage, the PCPG must be cooled by gas flow. For this purpose, the regular air flow is sustained through the holes (piezobrush® PZ2) or round slots (piezobrush® PZ3) integrated in the needle holder. The high voltage, transferred over plasma bridge to the needle electrode, generates an electric field sufficient to ignite a corona type plasma [211,212] to the electrically insulating objects, and a spark type plasma [213,214] to the electrically conducting surfaces. Figure 18b shows such a plasma between the needle tip and the grounded aluminum plate. The properties of the needle plasma depend on the distance to the substrate, the gap between the PCPG and the needle head, the radius of the needle tip, and the type of the substrate, especially its electric conductivity. These influences will be now considered more closely.

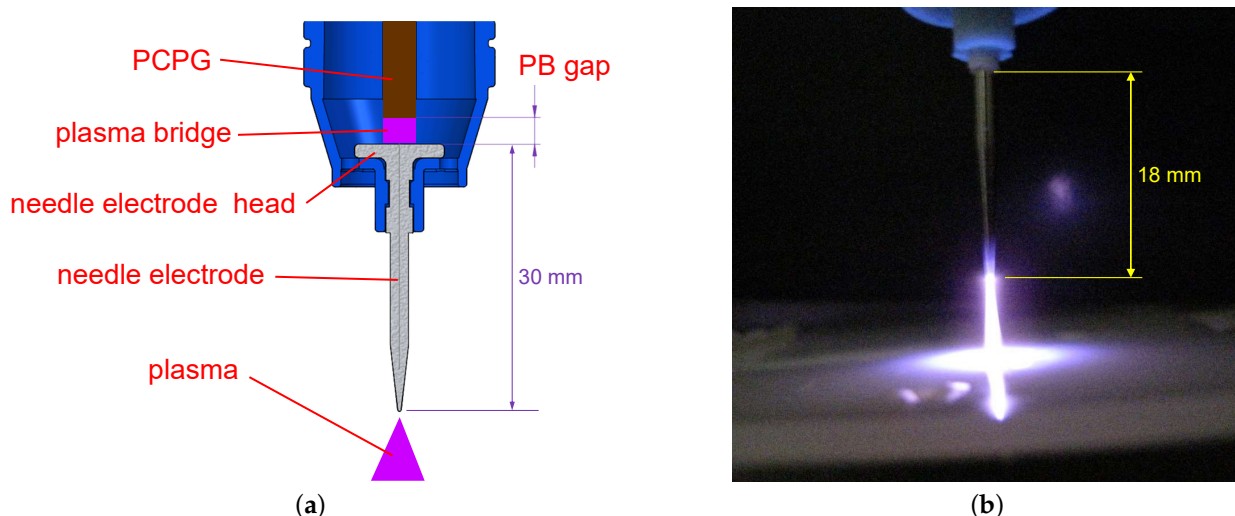

(**a**) (**b**)

**Figure 18.** The piezobrush® PZ3 needle nozzle. (**a**) discharge architecture; (**b**) spark discharge between the needle and the aluminum substrate.

### 7.1.1. Influence of the Distance

Figure 19a shows the dependence of the activation area on the distance between the needle tip and a 2 mm thick HDPE substrate placed on an aluminum plate. The two curves represent the results for two gaps between the PCPG and the needle head of 1 mm and 3 mm, respectively. For both gap sizes, the activation area is strongly increasing with decreasing distance. Since the shape of the activation area produced by the needle nozzle is circular, we can calculate the activation diameter, which reaches 18 mm for maximum power of 8.0 W. To obtain a more localized treatment, the PCPG power can be reduced or the distance can be increased. With decreasing gap, the activation area increases. The nozzle with the 1 mm gap reaches the activation areas about 30% larger than the nozzle with 3 mm gap. The bottom limitation of the gap size is the reduced cooling of the PCPG tip and the tolerances of the construction.

### 7.1.2. Influence of the Needle Tip Radius

The influence of the needle tip radius on the activation area is demonstrated in Figure 19b. In almost the entire commonly investigated distance range, the activation area for the sharp needle is larger than for the rounded one. The largest difference of 37% is reached for small distances. At distances of more than 17 mm, no significant difference

between activation areas for both needle tip radii can be observed. For a very sharp needle tip with a radius below 0.1 mm, the current flowing to the substrate heats up the needle tip strongly, causing an accelerated oxidation or even melting. Therefore, the needle radius of 0.2 mm is applied, as a compromise between the thermal overload of the needle tip and the less intensive discharge.

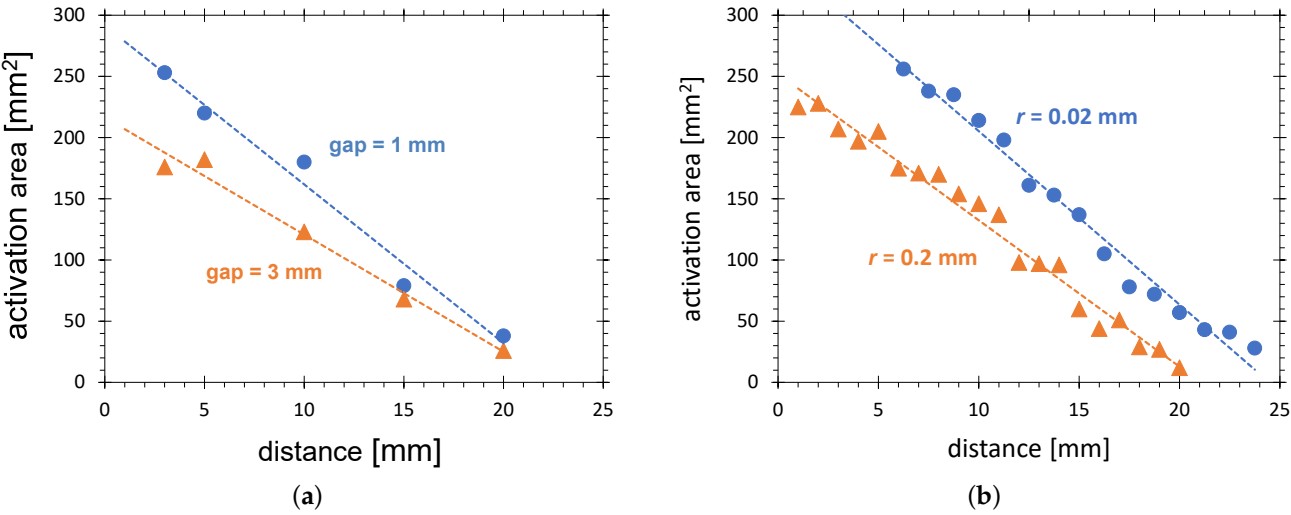

**Figure 19.** The dependence of the activation area on the distance between the needle tip and the substrate surface: (**a**) for two sizes of the gap between the PCPG and the needle head and (**b**) for two needle tip radii. During treatment, the piezobrush® PZ3 needle nozzle was operated in air with the PCPG power of 8.0 W. The 2 mm thick HDPE substrate is placed on a grounded aluminum plate.

### 7.2. Example: Treatment of Zirconia Dental Implants

Zirconia (zirconium dioxide) has recently gained importance as an implant material, replacing the standard metal-based materials like titanium in dentistry [215,216]. The technological breakthrough was possible by the application of high efficiency ceramics based on zirconia, such as Y-TZP (yttrium oxide stabilized tetragonal zirconia polycrystals). Especially successful in dentistry are so-called 3Y-TZP and 5Y-TZP containing 3% and 5% of $Y_2O_3$, respectively. These modern materials combine the traditional advantages of ceramics: chemical resistance and low thermal expansion, with properties typical for strong metals: mechanical strength and ductility. The zirconia-based materials exhibit strong hydrophobicity. It makes it difficult to join it with other materials. This is why the atmospheric pressure plasma has been applied for improvement of wettability and consequently of bond strength of Y-TZP-surfaces [217].

The broadly used term "Zirkon" for material of dental implants is from the chemical point of view the yttrium oxide stabilized tetragonal zirconia polycrystals (Y-TZP). Besides the wettability itself, the hydrophobic recovery of the wetting property is an important factor describing the result of the plasma treatment. The hydrophobic recovery can be evaluated by the time after which the decrease of the contact angle can be observed. The results of plasma treatment of plates and typical implants are shown in the following sections.

### 7.2.1. Treatment of Y-TZP Plates

A plasma treatment study on zirconia plates has been performed [218]. The substrates have been treated by the use of piezobrush® PZ2 with standard nozzle and with a needle nozzle (see the next page). The treatment time was 30 s. The nozzle distance from the substrate was 2 mm. For the standard nozzle, only very moderate improvement of wetta-

bility could be reached. The treatment with a needle nozzle is shown to be very efficient, resulting in a water contact angle of 14°, compared with 40° without treatment.

### 7.2.2. Plasma Needle Treatment of Y-TZP Implants

Figure 20a shows the plasma treatment by the use of a plasma-needle. The treatment was performed manually. The implant is rotated slowly around its axis, and the surface is plasma-brushed in the axial direction. The distance between the Y-TZP surface and the tip of the plasma-needle was 2 mm. No mechanical change, especially no roughening of the Y-TZP surface, can be observed.

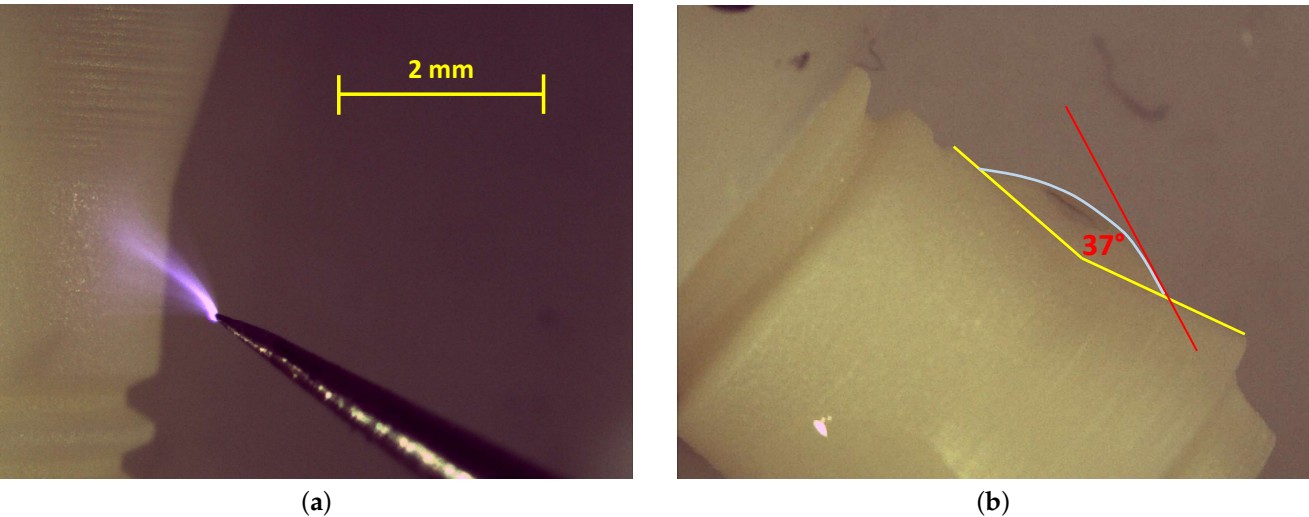

(**a**)  (**b**)

**Figure 20.** Y-TZP implant surface. (**a**) plasma needle treatment, and (**b**) graphical determination of the contact angle from the microscopic picture of the water droplet. The contours of the implant and of the droplet are traced for better visibility.

The method for determination of the contact angle is shown in Figure 20b. Because of the small area available for the droplet test, the contact angle of the implant is determined on the basis of microscopic pictures. The contact angle for water on the untreated surface is 37°. According to Ref. [219], the untreated ytrium stabilized Y-TZP surface shows the contact angle in the range from 66° to 74°.

The test droplet positioned at the implant surface immediately after treatment spreads in a very thin water layer. The measurement of the contact angle of such droplet is not possible. Excellent wetting is reached. According to the earlier studies, the improvement of wettability of Y-TZP surfaces is due to the polar component of the surface energy increase and correlates with an increase of oxygen and decrease of carbon concentration, as measured by XPS [217].

Y-TZP is known for strong hydrophobic recovery [220], which can be confirmed by our measurements. In Table 3, the wettability as a function of the time elapsed after treatments with two duration times is summarized. After one hour after 1 min plasma treatment, the contact angle is growing from about 0° to 17°. Four hours after one minute of treatment, the contact angle increases to 25°. Twenty-four hours after one minute of treatment with air plasma, the contact angle increases to about 35°. This corresponds to the contact angle before plasma treatment (complete hydrophobic recovery).

**Table 3.** Hydrophobic recovery: Influence of the storage time on the contact angle.

| Storage Time | 1 min Treatment | 5 min Treatment |
|:---:|:---:|:---:|
| 10 min | flat spreading | flat spreading |
| 30 min | flat spreading | flat spreading |
| 1 h | 17° | flat spreading |
| 2 h | | flat spreading |
| 3 h | | 26° |
| 4 h | 25° | |
| 19 h | | 35° |
| 24 h | 35° | |

To reduce the hydrophobic recovery, the treatment time with a plasma-needle was prolonged up to 5 min. It can be concluded that the 5 min treatment prolongs the onset of measurable hydrophobic recovery up to 2 h, comparing with less than 1 h after 1 min treatment. 3 h after 5 min treatment, the contact angle increases to 26°. 19 h after 5 min treatment, the contact angle increases to 35°. The complete hydrophobic recovery has been reached.

In conclusion, the wetting result delivered after treatment with piezobrush® PZ2 and standard nozzle is not sufficient. An excellent wettability has been already achieved after brief treatment (1 min) in environmental air with the piezobrush® PZ2 quipped with needle-nozzle. The onset of hydrophobic recovery after less than 1 h can be observed. After longer treatment (5 min) the onset of hydrophobic recovery occurs after more than 2 h. This time can be sufficient to perform the next process step.

## 8. Multi-Gas Nozzle

### 8.1. Operation Principle

The basic operation principle of the multi-gas nozzle (MGN) is the plasma generation on the tip of the needle described in Section 7.1 but in a flow of different gases, as shown in Figure 21a. Figure 21b shows the piezobrush® PZ2 with MGN treating water with argon plasma. This type of discharge is equivalent to the APPJ of the "plasma needle" type [221] or kINPen [222] (see the medical applications in 8.4). The most frequently used ionization gases are the noble gases, especially He or Ar, and their mixtures with diatomic gases ($O_2$,$N_2$ or $H_2$). However, mixtures with other gases can also be added, for example, water vapor, monomer vapors, hydrocarbons, fluorocarbons, siloxanes, and many others.

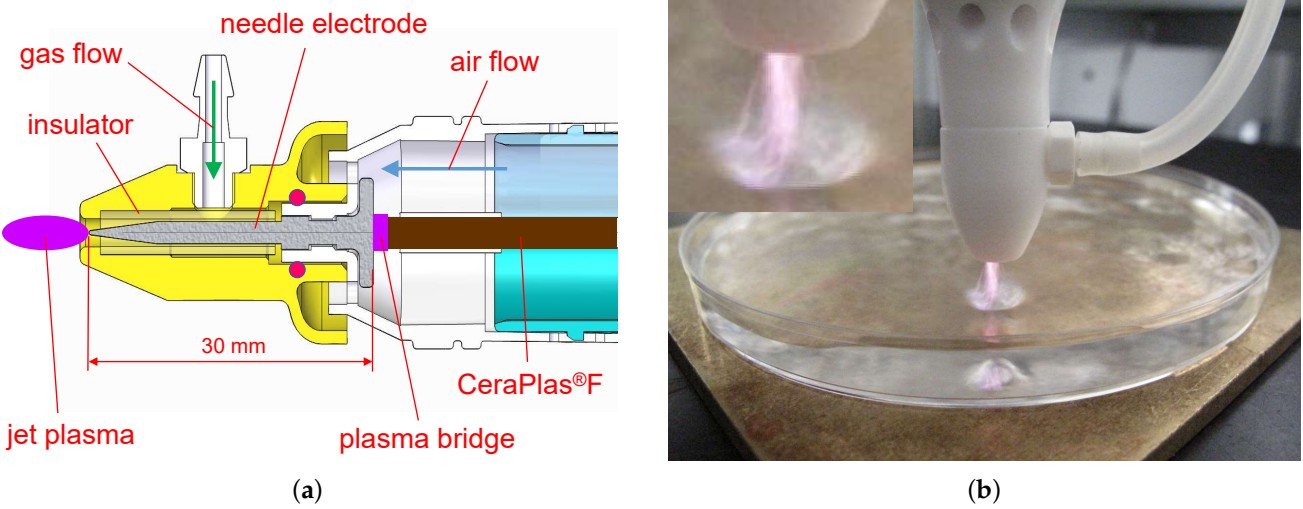

(**a**)  (**b**)

**Figure 21.** The multi-gas nozzle (**a**) operation principle, and (**b**) of the piezobrush® PZ2 treating water.

### 8.2. Activation Characteristics

For characterization of the activation performance of the MGN, argon was used. Figure 22 shows the activation area of piezobrush® PZ3 multi-gas nozzle for argon as a function of distance, and treatment time. The maximum activation area of 374 mm² reached on HDPE substrate at a distance of 3 mm from the needle tip is only slightly lower than the activation area obtained with air and open nozzle (compare with Figure 8b). The activation area decreases on average for a 17 mm² per 1 mm of distance increase, reaching about 100 mm² at a distance of 20 mm. This value is much higher than for treatment with air and an open nozzle. Two reasons for this MGN performance can be considered as most important: (i) the cross-section of the MGN orifice of only few mm² is on an order of magnitude smaller than for the open nozzle, resulting, even with much lower gas flow, in much higher gas speed, transferring the chemically active plasma products to the more remote points, and (ii) argon plasma produces long living metastable excited species (40 s [223]), which can carry the energy needed for chemical reactions in ambient air on considerably long distances. Activation area increases with activation time. The curve showing this dependence in Figure 22b has a general shape similar to the curve for surface activation with open nozzle and air (see Figure 9a). The three mechanisms explaining the saturation tendency for higher treatment time listed in Section 5.1.1 are valid also for the MGN treatment time dependence.

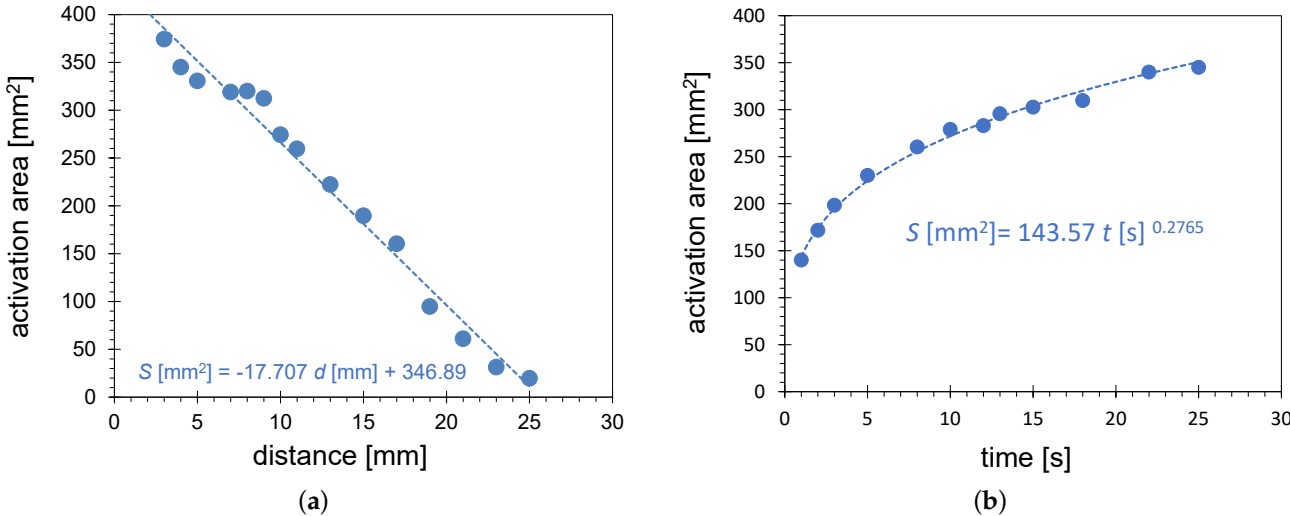

**Figure 22.** The activation area of piezobrush® PZ3 multi-gas nozzle for argon as a function of (**a**) distance, and (**b**) treatment time. Power level: 8 W, gas flow: 3.0 SLM, treatment time: 10 s. Visualization of activation area with 58 mN/m test ink on HDPE.

### 8.3. piezobrush® PZ2 Contra piezobrush® PZ1

The performance of the MGN of the piezobrush® PZ2 can be compared with piezobrush® PZ1, which is operated only with noble gases. The helium results of this comparison are displayed in Figure 23. Despite of a much lower input power of only 40% of the piezobrush® PZ1 input power and of the energy losses in the plasma bridge, the piezobrush® PZ2 operated with the MGN shows much better performance than the piezobrush® PZ1. The activation area reached by piezobrush® PZ2 with MGN is a factor of two larger than for the piezobrush® PZ1 (see Figure 23a). For reaching such a result, the MGN uses only a quarter of helium needed by piezobrush® PZ1. The operation gas flow ranges from 1 to 5 SLM for piezobrush® PZ2 and from 8 to 19 SLM for piezobrush® PZ1 shown in Figure 23b are not overlapping, because the RPT in the piezobrush® PZ1 requires the comparatively high minimum gas flow for cooling. This is not required for MGN, because the CeraPlas™ F used in piezobrush® PZ2 is cooled by air. Consequently, the gas

flow optimal for the activation process of about 3 NLM can be applied. The main reason for such a big difference in performance are the piezoelectric devices used. The RPT used in the piezobrush® PZ1 has much lower voltage transformation ratio and much higher power loss ratio than the CeraPlas™ F.

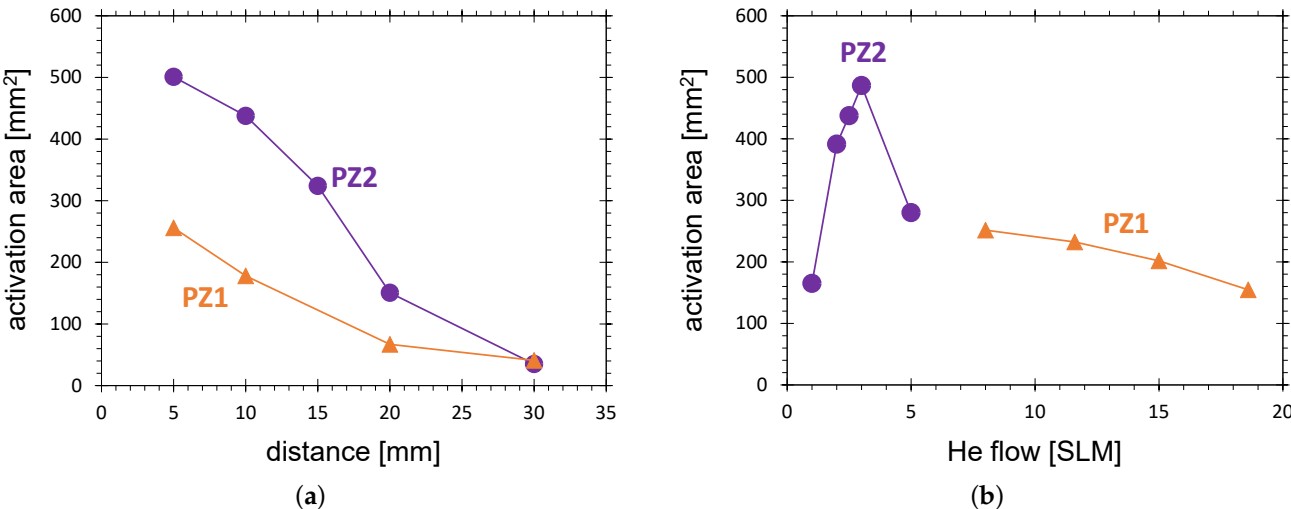

**Figure 23.** Comparison of the piezobrush® PZ2 and piezobrush® PZ1 activation area after 10 s treatment with helium plasma, visualized on HDPE with test ink gauged for 58 mJ/m², as a function of (**a**) distance between the PCPT and the substrate (gas flow of the piezobrush® PZ1 and piezobrush® PZ2 is 10 SLM and 2.5 SLM, respectively), and (**b**) helium flow (distance of the piezobrush® PZ1 and piezobrush® PZ2 from the substrate is 5.5 mm). The piezobrush® PZ1 and piezobrush® PZ2 were operated with 20 W and 8.3 W input power, respectively.

### 8.4. Potential for Medical Applications

The arc-free atmospheric pressure cold plasma jets [224,225] constitute the largest family of cold atmospheric plasma (CAP) devices used for inactivation of microorganisms [226] and other medical applications [227,228]. The representative examples of such devices are the so-called plasma needle [221], the plasma pencil [229], and the kINPen [222]. The applications of the plasma needle and the kINPen are reviewed shortly, because they can be implemented by use of the MGN.

#### 8.4.1. Plasma Needle

The plasma needle was developed for a surface treatment of biological materials [221]. It is a small device consisting of a needle electrode with a diameter of 0.3 mm, separated from the cylindrical grounded electrode by a dielectric barrier. It typically works with He or other noble gases. Standard operating parameters are the flow of 2 SLM and the rf (13.56 MHz) power of 0.5–2 W. It was used for the treatment of cultured cells [230,231]—especially for bacterial inactivation [232], treatment of dental cavities [233,234], deactivation of *Escherichia coli* [235], and *Streptococcus mutans* bacteria [236]. The treatment of mammalian cells (reattachment and apoptosis achieved) [237,238] was also conducted in vivo [239,240].

#### 8.4.2. The KINPen

The kINPen, similarly to the plasma needle, consists of a pin-type electrode with a diameter of 1 mm surrounded by a quartz capillary with an inner diameter of 1.6 mm and an outer grounded electrode. It is also used with noble gases, typically argon. The voltage of the 1.1 MHz signal supplied ranges from 2 to 6 kV. The coupled power is 2–3 W. Thanks to the very low thermal load of less than 150 mW, it is very suitable for tissue studies [241], such as in risk assessments of the application of a plasma jet in dermatology [242]. It was tested for blood coagulation [243], for wound healing [244], and for the destruction of

malignant melanoma [245], colon cancer cells [246], and pancreatic cancer cells in vitro and in vivo [247]. It shows antimicrobial properties [248]. However, kINPen is designed for the very low power level needed for the treatment of living tissue, not for the effective inactivation of microorganisms.

## 9. Conclusions and Outlook

This review shows the use of three types of PCPG for generation of the PDD. Based on the PCPG, several handheld, versatile plasma tools were developed, allowing the treatment of wide range of thermally sensitive substrates such as implants, fruits, or liquids. This versatility is achieved by using different excitation structures, such as DBD, APPJ, FE-DBD, SMD, or plasma needle, all powered by the same PCPG.

PCPG is able to produce a high ozone concentration in air or in oxygen. The ozone concentrations of 250 in air and 800 ppm in oxygen, and the production rates of 80 mg/h and 250 mg/h, respectively, are achieved.

For evaluation of the activation performance of the different discharge configurations, the activation area on LDPE substrates visualized by use of 58 mN/m test inks is used. For both: open nozzle and DBD nozzle operation the activation results correlate with the number of micro-discharges per PCPG oscillation cycle.

The correlation between the specific discharge architecture and its optimal processing targets is discussed. Specifically, three configurations of PCPG driven DBD discharges are evaluated on the basis of activation area. The best results were achieved with single DBD with the excitation electrode driven over the plasma bridge. The second best is the configuration with the double DBD driven directly by the PCPG. The weakest results were achieved with double DBD with the excitation electrode driven over the plasma bridge.

The needle electrode powered by PCPG over the plasma bridge is able to produce plasma in different gases, reaching the activation results comparable with the open PDD.

The generation of piezoelectric direct discharge is a considerably new discipline in the atmospheric pressure plasma world. Consequently, many interesting questions are still not answered. Some examples of scientifically challenging subjects are:

- Physics of the PDD plasma bridge, especially its temporal development, and electric parameter determining its power coupling capacity.
- Influence of humidity on the PDD properties, chemistry, and microbiocidal activity.
- Control of the PDD chemistry by shaping the excitation signal, for example by pulse width modulation.

**Funding:** This research received no external funding.

**Informed Consent Statement:** Not applicable.

**Data Availability Statement:** The data presented in this study are available on request from the corresponding author.

**Acknowledgments:** The development of the piezobrush® PZ3 was supported by TDK Electronics GmbH. The experiments related to the treatment of the titanium plates and zirconia implants were conducted by Ute Schmidt.

**Conflicts of Interest:** The authors declare no conflict of interest.

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
