# Peer review of "Piezoelectric Direct Discharge: Devices and Applications"

_plasma, doi:10.3390/plasma4010001_

Round 1

Reviewer 1 Report

This article presents an extensive review on plasmas generated by resonant piezoelectric transformer, also denomined as Piezoelectric Direct Discharge. The review is well written and presents to the reader a wide range of content in order to undestand this peculiar type of plasma technology and its applications. So, it is recommended the publication of the review in "Plasmas" journal.

In order to improve paper quality some points are addressed:

(1) In Figure 2, please put the frequency in kHz.

(2) Figure 3, please put a scale in figures. It is important to know the dimensions of the devices. It is important to put the citation in the Figure 3 (Also, for other Figures in the article).

(3) Figure 4, please consider the unit in microseconds.

(4) Figure 12, please insert a scale in figures. Also for Figures 13, 17, 18, 20a and 21.

(5) Yet for Figure 20, it is not clear where is the water dropet in Fig, 21b, please improve this image. 

Author Response

The response to the criticism:

This article presents an extensive review on plasmas generated by resonant piezoelectric transformer, also denominated as Piezoelectric Direct Discharge. The review is well written and presents to the reader a wide range of content in order to understand this peculiar type of plasma technology and its applications. So, it is recommended the publication of the review in "Plasmas" journal.

In order to improve paper quality some points are addressed:

(1) In Figure 2, please put the frequency in kHz.

 The units on the x-axis are changed to kHz.

(2) Figure 3, please put a scale in figures. It is important to know the dimensions of the devices.

The reference sizes are added in Figures 3a and 3b

 It is important to put the citation in the Figure 3

The pictures in Figure 3 are taken by the first author.

(Also, for other Figures in the article).

The citations in figure captions are added in these cases when the content
is not published first in this review.

(3) Figure 4, please consider the unit in microseconds.

The units are changed to microseconds.

(4) Figure 12, please insert a scale in figures.

The reference size is added.

Also, for Figures 13, 17, 18, 20a and 21.

The reference sizes are added in Figures 13, 17, 18, 21.
The scale is added in Figure 20a.

(5) Yet for Figure 20, it is not clear where is the water droplet

The contour of the implant and of the droplet
are traced for better visibility.

in Fig, 21b, please improve this image. 

The magnification of the discharge is added.

Reviewer 2 Report

What is the type of PZT utilized?  Is it hard or soft or the type?

Figure 2, the legend should be explained with symbols

Line 302: spelling mistake: capcitive

Figure 22, equations of a line written as y=f(x), what is x and what is y?  Use the symbols of the graph instead (units too).

Is the research supported by a company? The company that sells the brushes?

Author Response

The response to the criticism:

What is the type of PZT utilized?  Is it hard or soft or the type?

The “high stiffness” is added to the information about PT in section 2.1.4.

Figure 2, the legend should be explained with symbols

The symbol zeta is added to the curve description.

Line 302: spelling mistake: capcitive

corrected

Figure 22, equations of a line written as y=f(x), what is x and what is y?  Use the symbols of the graph instead (units too).

 The equations are written as S=f(d) and S=f(t). The units are added

Is the research supported by a company? The company that sells the brushes?

The research is conducted by the company, which is the affiliation of all three authors:
relyon plasma GmbH. This is the developer and producer of the piezobrushes PZ1, PZ2 and PZ3 referred in the review. The sales are made mainly over sales network.
Currently, the relyon plasma GmbH is 100% company of TDK, the developer and producer of the CeraPlas piezoelectric transformers. In the time of the development of the PZ1 and PZ2, the TDK was not involved financially in relyon plasma GmbH. During the development of PZ3, the relyon plasma was 50% TDK company (support disclosed in acknowledgement).